# Identification of common predisposing loci to hematopoietic cancers in four dog breeds

**Benoît Hédan** [1]*, **Édouard Cadieu**[1], **Maud Rimbault**[1], **Amaury Vaysse**[1], **Caroline Dufaure de Citres**[2], **Patrick Devauchelle**[3], **Nadine Botherel**[1], **Jérôme Abadie**[4], **Pascale Quignon**[1], **Thomas Derrien**[1], **Catherine André**[1]

**1** Univ Rennes, CNRS, IGDR (Institut de Génétique et Développement de Rennes)–UMR6290, Rennes, France, **2** Antagene, La Tour-de-Salvagny, France, **3** Micen Vet, Créteil, France, **4** Oniris, Laboniris—Department of Biology, Pathology and Food Sciences, Nantes, France

* benoit.hedan@univ-rennes1.fr

**Data Availability Statement:** All genotyping data ia available at doi:10.5061/dryad.hx3ffbgd4.

## Abstract

Histiocytic sarcoma (HS) is a rare but aggressive cancer in both humans and dogs. The spontaneous canine model, which has clinical, epidemiological, and histological similarities with human HS and specific breed predispositions, provides a unique opportunity to unravel the genetic basis of this cancer. In this study, we aimed to identify germline risk factors associated with the development of HS in canine-predisposed breeds. We used a methodology that combined several genome-wide association studies in a multi-breed and multi-cancer approach as well as targeted next-generation sequencing, and imputation We combined several dog breeds (Bernese mountain dogs, Rottweilers, flat-coated retrievers, and golden retrievers), and three hematopoietic cancers (HS, lymphoma, and mast cell tumor). Results showed that we not only refined the previously identified HS risk *CDKN2A* locus, but also identified new loci on canine chromosomes 2, 5, 14, and 20. Capture and targeted sequencing of specific loci suggested the existence of regulatory variants in non-coding regions and methylation mechanisms linked to risk haplotypes, which lead to strong cancer predisposition in specific dog breeds. We also showed that these canine cancer predisposing loci appeared to be due to the additive effect of several risk haplotypes involved in other hematopoietic cancers such as lymphoma or mast cell tumors as well. This illustrates the pleiotropic nature of these canine cancer loci as observed in human oncology, thereby reinforcing the interest of predisposed dog breeds to study cancer initiation and progression.

## Author summary

Because of specific breed structures and artificial selection, pet dogs suffer from numerous genetic diseases, including cancers and represent a unique spontaneous model of human cancers. Here, we focused on histiocytic sarcoma (HS), a rare and highly aggressive cancer in humans. In this study, we have used spontaneous affected and unaffected dogs from three predisposed dog breeds to identify loci involved in HS predisposition. Through genetic analyses, we showed that these canine cancer predispositions are due to the additive effect of several risk haplotypes also involved in the predisposition of other

**Funding:** CA received fundings from INCa PLBio (https://www.e-cancer.fr/Institut-national-du-cancer/) (Grant "canine rare tumours" funding (N° 2012-103; 2012-2016) and Aviesan (https://aviesan.fr/) (Grant MTS 2012-06) for the work describe here. BH received fundings from American Kennel Club Canine Health fundation (https://www.akcchf.org/) (Grant N 2446). This research is also funded by ANR (Grant ANR-11-INBS-0003). The funders had no role in study design, data collection and analysis, decision to publish, or preparation of the manuscript.

**Competing interests:** The authors have declared that no competing interests exist.

hematopoietic cancers. The corresponding chromosomal regions in humans are involved in the predisposition of several cancers and are also associated with immune traits. This study demonstrates the pleiotropic nature of these canine cancer loci as observed in human oncology, thereby reinforcing the interest of predisposed dog breeds to study mechanisms involved in cancer initiation.

## 1. Introduction

Over the past decade, dogs have emerged as a relevant and under-used spontaneous model for the analysis of cancer predisposition and progression as well as development and trials of more efficient therapies for many human cancers [1–10]. With over 4.2 million dogs diagnosed with cancer annually in the USA [8], canine cancers represent a unique source of spontaneous tumors. Canine cancers share strong similarities with human cancers, based on both the biological behavior and histopathological features [11–14]. Thus, spontaneous canine models are a natural and ethical model i.e., a non-experimental model to decipher the genetic basis of cancers. Given the incomplete penetrance and genetic heterogeneity of human cancers, identifying their genetic predisposition is complex [6], and almost impossible in rare cancers. Because of specific breed structures and artificial selection, dog breeds have gained numerous susceptibilities to genetic diseases, and a limited number of their critical genes are involved in complex diseases such as cancers [6]. Further, numerous genome-wide association studies (GWAS) in dogs have illustrated that with complex traits, such as body size and cancer, a small number of loci with strong effects are involved in dogs, as compared to humans, thereby facilitating their identification. With large intra-breed linkage disequilibrium (LD), cancer loci have been successfully identified even in studies with a small number of cases and controls [15–20]. Thus, spontaneously affected pet dogs, with breed-specific cancers, provide efficient natural models to identify the genetics underlying several dog-human homologous cancers.

Histiocytic sarcoma (HS), which involves histiocytic cells (dendritic or monocytic/macrophagic lineages), is extremely rare in humans, and is associated with a limited response to chemotherapy and high mortality. Due to the rarity of this cancer, there is no consensus on its prognostic factors and standard treatment [21]; therefore, models are urgently needed to better understand this aggressive cancer. In the entire dog species, HS is also a relatively rare cancer; however, a few popular breeds, such as Bernese mountain dogs (BMD), Rottweilers, retrievers (especially flat coated retrievers [FCR]), are highly predisposed to this cancer with breed-specific clinical presentations. Interestingly, the clinical presentation and histopathology of this canine cancer are similar to those observed in humans [22,23]. These breed-specific predispositions have allowed to sample numerous cases and led to the successful identification of shared somatic mutations between human and canine HS involving the mitogen-activated protein kinase (MAPK) pathway [24,25]. We recently showed that the same mutations of protein tyrosine phosphatase non-receptor type 11 (*PTPN11*), the most frequently altered gene of the MAPK pathway, are found in both human and canine HS [25]. Most importantly, thanks to these breed predispositions, we have previously shown that somatic mutations of *PTPN11* found in half of the HS canine cases are linked to an aggressive HS clinical subgroup in both dogs and humans [25]. Regarding predisposition of HS, a previous study with 236 cases and 228 controls has highlighted that S-methyl-5′-thioadenosine phosphorylase (*MTAP*)—cyclin-dependent kinase inhibitor 2A (*CDKN2A*) genomic region is one of the main loci that confers susceptibility to HS in BMD [26]. Nevertheless, HS is a multifactorial cancer, and other loci are expected to be involved in HS predisposition. This is in accordance with the fact that,

despite the awareness and attempts to select against HS for 20 years, breeders have not succeeded in reducing the prevalence of this devastating cancer because of its strong heritability in BMD [27]. In addition, it is suspected that HS-predisposed breeds (BMD, Rottweiler, and retrievers) share common risk alleles due to common ancestors; thus, cases from close breeds can accelerate the identification of common loci by reducing the haplotype of these critical regions [28]. Finally, it is worth noting that these HS-predisposed breeds also present a high risk of developing other cancers such as lymphomas, mast cell tumors, hemangiosarcomas, osteosarcomas, or melanomas [26,29,30]. It is estimated that a high proportion of deaths in these HS-predisposed breeds are due to several neoplasms (BMD: 45–76%, golden retriever: 39–50%,Labrador retriever: 31–34%,FCR: 54%, and Rottweiler: 30–45%) [30–32].

This study aimed to extend previous studies by deciphering the genetic basis of HS based on a multi-breed approach. We performed exhaustive GWAS with a substantially increased numbers of cases and controls from three different breeds, and with higher density single nucleotide variation (SNV) arrays. Our results not only strengthen the crucial role of the *CDKN2A* locus in HS, but also shed light on secondary loci located on canine chromosomes 2, 5, 14 and 20 containing relevant novel candidate genes. They point toward the existence of regulatory variants in non-coding regions and/or methylation mechanisms linked to risk haplotypes, which ultimately lead to strong cancer predisposition in specific dog breeds.

## 2. Results

To decipher the genetic basis of HS in the dog model system, we took advantage of the well-known HS-predisposed breeds. We combined data from GWAS with high-density genotyped and imputed SNV data from BMD, FCR, and Rottweiler with HS, lymphomas, and mast cell tumors as well as publicly available data from lymphoma and mast cell tumors in golden retrievers [15,16] (Table 1).

### 2.1. Identification of loci linked to HS risk development in BMD breed

Using BMD DNA from 172 HS cases and 128 controls, we performed the first round of GWAS (GWAS_1_HS_BMD) by correcting for population stratification and cryptic relatedness (Fig 1). From 10,3487 SNVs left after applying filters, we identified 21 SNVs that were significantly associated with HS, including 20 SNVs on chromosome 11 spanning 40.3–47.2 Mb (strongest associated SNV was CFA11:41,161,441, $p_{corrected} = 3.11 \times 10^{-7}$) and one SNV on chromosome 20 (CFA20:30,922,308, $p_{corrected} = 3.73 \times 10^{-5}$). Moreover, an additional SNV on chromosome 5 (CFA5:30,496,048, $p_{corrected} = 9.48. \times 10^{-5}$) was close to the genome-wide significance, and was suspected to be associated with HS. This GWAS confirmed that the main locus linked to HS was located on CFA11, overlapping the *MTAP-CDKN2A* region, a locus previously associated with HS [26]. The analysis also identified a new locus on CFA20 and suggested the existence of another locus on CFA5. Interestingly, these three regions were previously identified for cancer predisposition in dogs: CFA11 (41.3–41.4 Mb) in osteosarcoma, CFA5 (29.6–34.1 Mb) in lymphomas and hemangiosarcomas, CFA20 (30.9–50.1 Mb) in mast cell tumors [15–17]. Indeed, Tonomura et al. has identified two independent peaks on CFA5 involved in lymphomas and hemangiosarcomas [16] overlapping the CFA5 locus suspected in HS; while, Arendt et al. has identified at least two independent peaks on CFA20 involved in mast cell predisposition [15], which also overlap the CFA20 locus found in HS in the BMD breed. Thus, we hypothesized that, because of strong breed selection, the significant associations detected for HS could be due to the cumulative risk alleles/haplotypes that can also be at risk for other hematopoietic cancers. This hypothesis was reinforced by the strong predisposition of BMD to hematopoietic cancers such as lymphomas [33,34] and by the fact that a

**Table 1. Characteristics of the genome-wide association studies (GWAS) analyses performed in this study.** * Dogs genotyped on the Affymetrix Axiom Canine Genotyping Array were also available in the Illumina 173K SNV Canine HD array format.

| GWAS Name | Paragrah | cancers | breeds | SNP arrays format | Number of cases after QC | Number of controls after QC |
|---|---|---|---|---|---|---|
| GWAS_1_HS_BMD | §2.1 | HS | BMD | Illumina 173k SNV Canine HD array | 172 | 128 |
| GWAS_2_HS+lymphoma_BMD | §2.2a | HS and lymphoma | BMD | Illumina 173k SNV Canine HD array | 252 | 128 |
| GWAS_3_HS+lymphoma_BMD +golden_retriever | §2.2a | HS and lymphoma | BMD, golden retriever | Illumina 173k SNV Canine HD array | 293 | 300 |
| GWAS_4_HS+MCT_BMD | §2.2b | HS and mast cell tumor | BMD | Illumina 173k SNV Canine HD array | 216 | 128 |
| GWAS_5_HS+MCT_BMD+golden_retriever | §2.2b | HS and mast cell tumor | BMD, golden retriever | Illumina 173k SNV Canine HD array | 285 | 202 |
| GWAS_6_HS_BMD_with_imputed_SNV | §2.3 | HS | BMD | Affymetrix Axiom Canine Genotyping array 1.1M SV (n = 113)* Illumina 20k SNV Canine HD array imputed for 1.1M SV (n = 464) Illumina 173k SNV Canine HD array imputed for 1.1M SV (n = 300) | 403 | 347 |
| GWAS_7_HS_BMD+FCR_with_imputed_SNV | §2.3 | HS | BMD, FCR | Affymetrix Axiom Canine Genotyping array 1.1M SV (n = 134)* Illumina 20k SNV Canine HD array imputed for 1.1M SV (n = 464) Illumina 173k SNV Canine HD array imputed for 1.1M SV (n = 328) | 416 | 362 |
| GWAS_8_HS_BMD+FCR +Rottweiler_with_imputed_SNV | §2.3 | HS | BMD, FCR, Rottweiler | Affymetrix Axiom Canine Genotyping array 1.1M SV (n = 134)* Illumina 20k SNV Canine HD array imputed for 1.1M SV (n = 464) Illumina 173k SNV Canine HD array imputed for 1.1M SV (n = 388) | 453 | 385 |

majority of BMDs (58.3% to 66.5%) succumb to cancer, in first place HS, lymphomas, or mast tumors [26,30,35]. Moreover, we observed that in HS-affected BMD families, relatives of HS affected dogs are frequently affected by other hematopoietic cancers such as mast cell tumors and lymphomas [36] (S1 Fig).

## 2.2. Involvement of HS loci in other hematopoietic cancers

To test whether these HS predisposing loci could also be involved in the predisposition of lymphomas or mast cell tumors, we added lymphoma or mast cell tumor cases to the previous HS GWAS.

**2.2.a. Lymphoma.** We performed a second GWAS (GWAS_2_HS+lymphoma_BMD) by adding 80 lymphoma-affected BMDs to the first GWAS (GWAS_1_HS_BMD). We identified six significantly associated SNVs, one of which was located on the CFA5 (best SNV on CFA5:30,496,048, $p_{corrected} = 5.88 \times 10^{-6}$; Figs 2 and S2). This result confirmed that the CFA5

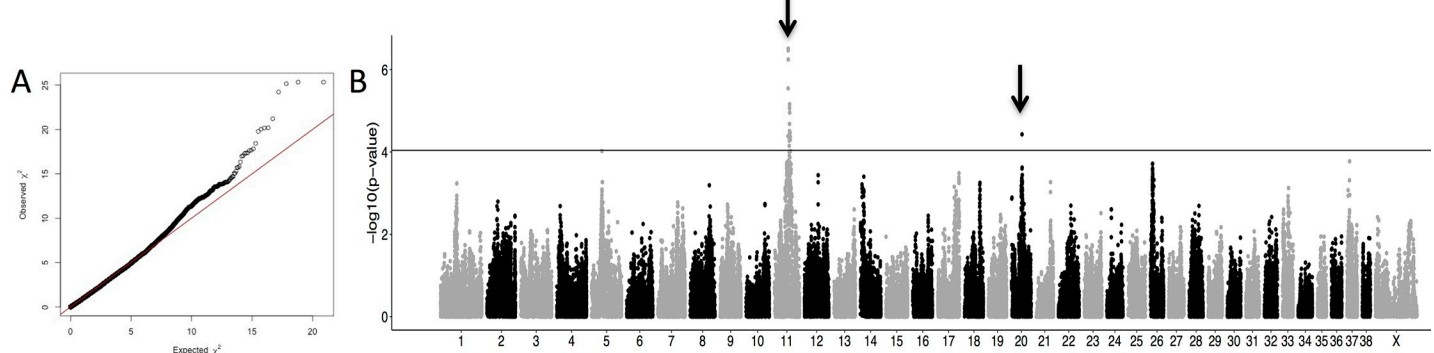

**Fig 1. Results of genome-wide association studies (GWAS) on Bernese mountain dogs (BMD) with 172 histiocytic sarcoma (HS) cases and 128 controls (GWAS_1_HS_BMD).** A) Quantile-quantile plot displaying a genomic inflation λ of 1.000005, indicating no residual inflation. B) Manhattan plot displaying the statistical results from the GWAS. This analysis pointed out two loci (arrows) on chromosome 11 (CFA11:41161441, $p_{corrected} = 3.11 \times 10^{-7}$) and on chromosome 20 (CFA20:30922308, $p_{corrected} = 3.73 \times 10^{-5}$).

locus is common to the predisposition of both HS and lymphoma in BMD. SNVs in LD ($R^2 >$ 0.6) with the top SNV of CFA5 delimited a locus from 28.3 Mb to 34.4 Mb (Fig 2). The top SNV of the CFA5 locus was located in an intron of the sphingolipid transporter 3 (*SPNS3*) gene for which the paralogous gene (*SPNS2*) is known to be important in immunological development, and inflammatory and autoimmune diseases [37]. Since these SNVs are located in one of the two lymphoma predisposing peaks (29.8 Mb and 33 Mb) found by Tonomura et al. in the golden retrievers, we performed a meta-analysis by including BMD GWAS data from this study and the publicly available golden retriever GWAS data [16], resulting in a final data set of 93,100 SNVs. This GWAS (GWAS_3_HS+lymphoma_BMD+golden_retriever), after adding golden retriever lymphoma cases (n = 41) and controls (n = 172) to the second BMD GWAS (GWAS_2_HS+lymphoma_BMD), contained 293 HS or lymphoma cases and 300 controls. We observed an increased signal on the CFA5 locus, thereby strengthening the two loci previously identified by Tonomura et al. at 29.8 Mb and 33 Mb (SNV CFA5:29,836,124, $p_{corrected} = 1.86 \times 10^{-6}$ and SNV CFA5:32,824,053, $p_{corrected} = 2.2 \times 10^{-7}$; Figs 2 and S2). These results show that BMDs and golden retrievers share common risk loci on CFA5 that is involved in hematopoietic cancers. Further, the CFA5 locus for both HS and lymphomas in BMD overlaps the two independent loci associated with lymphoma and hemangiosarcoma risk in golden retrievers (29.8 Mb and 33 Mb, respectively; Fig 2).

**2.2.b. Mast cell tumor.** By combining the first HS BMD GWAS (GWAS_1_HS_BMD) and 44 BMDs with mast cell tumors (GWAS_4_HS+MCT_BMD), we identified six significantly associated SNVs (best SNV on CFA11:41,161,441, $p_{corrected} = 6.93 \times 10^{-7}$), of which one was located on the CFA20 (best SNV on CFA20 CFA20:30,922,308, $p_{corrected} = 1.53 \times 10^{-5}$; Figs 3 and S3). This result confirmed that the CFA20 locus was common to HS and mast cell tumors (MCT) predisposition in BMD. SNVs in LD ($R^2 > 0.6$) showed that the top SNV of CFA20 delimited a locus from 29.3 Mb to 32.8 Mb (Fig 3). The top SNV of CFA20 locus lies in an intron of fragile histidine triad diadenosine triphosphatase (*FHIT*), a tumor suppressor involved in apoptosis and prevention of epithelial-mesenchymal transition [38]. Interestingly, this locus overlapped one of the three MCT-independent predisposing peaks (33 Mb, 39 Mb, and 45 Mb; canFam3) identified in golden retrievers [15]. We then performed a meta-analysis combining our BMD GWAS for HS and MCT (GWAS_4_HS+MCT_BMD) with the golden retriever GWAS for MCT by adding publicly available data [15] to create a final data set of 88,202 SNVs. The addition of European golden retriever MCT cases and controls resulted in

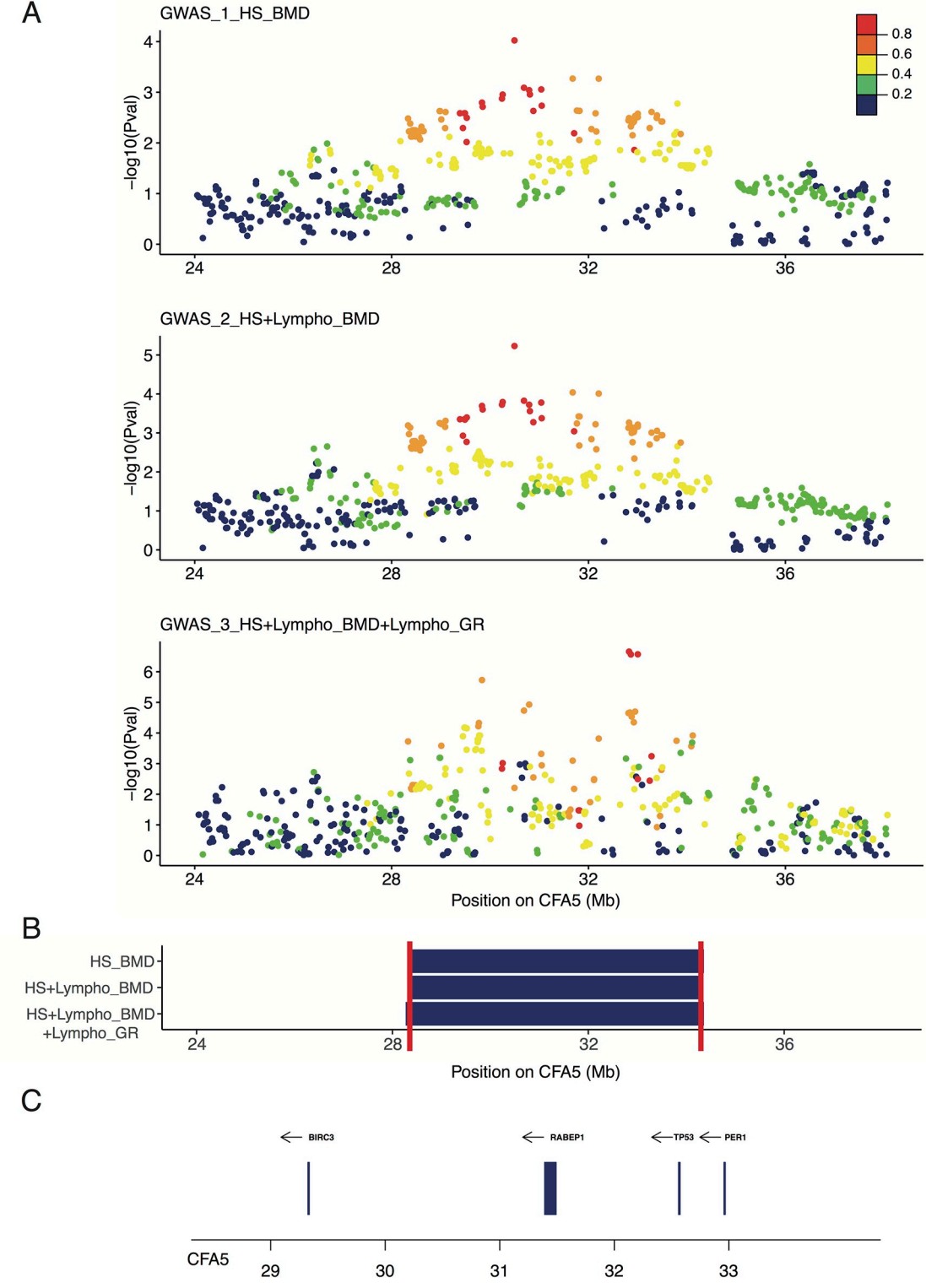

**Fig 2. Close up view of the CFA5 locus.** A) Manhattan plot of the CFA5 20–40 Mb region highlighting the best *p*-values obtained in the three genome-wide association studies (GWAS): Bernese mountain dog (BMD) GWAS for histiocytic sarcoma (HS) with 172 cases vs. 128 controls (GWAS_1_HS_BMD); BMD GWAS for HS and lymphoma with 252 cases vs. 128 controls (GWAS_2_HS +lymphoma_BMD); meta-analysis combining the BMD GWAS of HS and lymphoma (252 cases vs. 128 controls) and golden retriever GWAS for lymphoma (41 cases vs. 172 controls) from Tonomura et al. [16] (GWAS_3_HS+lymphoma_BMD +golden_retriever). The $R^2$ in cases from the top single nucleotide variation (SNV) is depicted to show the linkage-disequilibrium

(LD) structure. B) Regions delimitated by the SNPs in LD with the best GWAS SNVs ($R^2 > 0.6$) in cases; minimal region between the three GWAS (CFA5:28309815–34321500) is delimitated by red lines. C) Close up view of the genes (with available symbols) located in this minimal region (28–34 Mb) of CFA5.

an increased association signal in the CFA20 locus, and clearly pointed out the CFA20 locus at 33 Mb (best SNV on CFA20:33,321,282, $p_{corrected} = 4.79 \times 10^{-7}$) in rho guanine nucleotide exchange factor 3 (*ARHGEF3*) and close to interleukin 17 receptor D (*IL17RD*), one of the three peaks identified by Arendt et al. These results show that the BMD and golden retriever breeds share common inherited risk factors on CFA20 for HS and MCT (Figs 3 and S3). These results also confirmed that the association of CFA20 with MCT in the golden retrievers is due to the additional effect of at least three risk haplotypes (33 Mb, 39 Mb, and 45 Mb).

In conclusion, it appears that the loci linked to cancer in dogs can have pleiotropic effects associated with the risk of several cancers in several breeds, and can be due to the additive effects of several risk peaks. Thus, to more precisely identify the HS risk haplotypes and reduce the loci size, we added new HS cases from BMD and other HS predisposed breeds to the first BMD GWAS.

## 2.3. Refining HS loci by multiple-breed analyses and imputation on higher density SNV array

To increase the power of the GWAS and refine the HS loci, we added HS cases and controls from FCR and Rottweiler breeds genotyped on Illumina 173K SNV Canine HD. To increase the density of markers, 134 dogs (113 BMDs and 21 FCRs) were re-genotyped on the higher density Affymetrix Axiome Canine Genotyping array (1.1M SNV), and were used as a reference panel to impute these SNVs on the Illumina 173K SNV Canine HD. In addition, we also added data from previously published BMD cases and controls [26] by imputing their genotypes from the Canine SNP20 Bead-Chip panel (Illumina -22K SNV) to the higher density Axiome Canine Genotyping array (1.1M SNV). The quality of imputation was evaluated by masking imputed SNVs on half of the BMDs genotyped on the high-density Affymetrix Axiome Canine Genotyping array. The mean concordances between the masked autosomal SNVs and imputed SNVs were 91.97% and 95.86% for SNVs imputed from the Illumina -22k SNV panel to the higher density Affymetrix Axiome Canine Genotyping array and from the Illumina 173K SNV Canine HD panel to the higher density Affymetrix Axiome Canine Genotyping array, respectively. These concordances are similar to those described by the work of Friedenberg and Meurs, which describes a genotype concordance of up to 92.4% with Beagle software [39].

The addition of BMD cases and controls to the first BMD GWAS (GWAS_1_HS_BMD) resulted in a total of 403 cases and 347 controls imputed to form a final data set of 488,872 SNVs (GWAS_6_HS_BMD_with_imputed_SNV). Statistical analysis allowed the identification of 1,730 SNVs significantly associated with HS (Table 2, Fig 4A and 4B). This GWAS, after increasing the number of BMD cases and controls, confirmed the involvement of the CFA11 locus as well as the role of other loci (CFA5 and CFA14), and identified a new locus on chromosome 2 in HS BMD predisposition.

By the addition of 28 FCRs (13 cases and 15 controls) to GWAS_6_HS_BMD_with_imputed_SNV to form a final dataset of 525,657 SNVs (GWAS_7_HS_BMD+-FCR_with_imputed_SNV), we identified 1,558 SNVs that were significantly associated with HS (Table 2, Fig 4C and 4D), confirming the involvement of the CFA2, CFA5, CFA11, and CFA14 loci in HS predisposition.

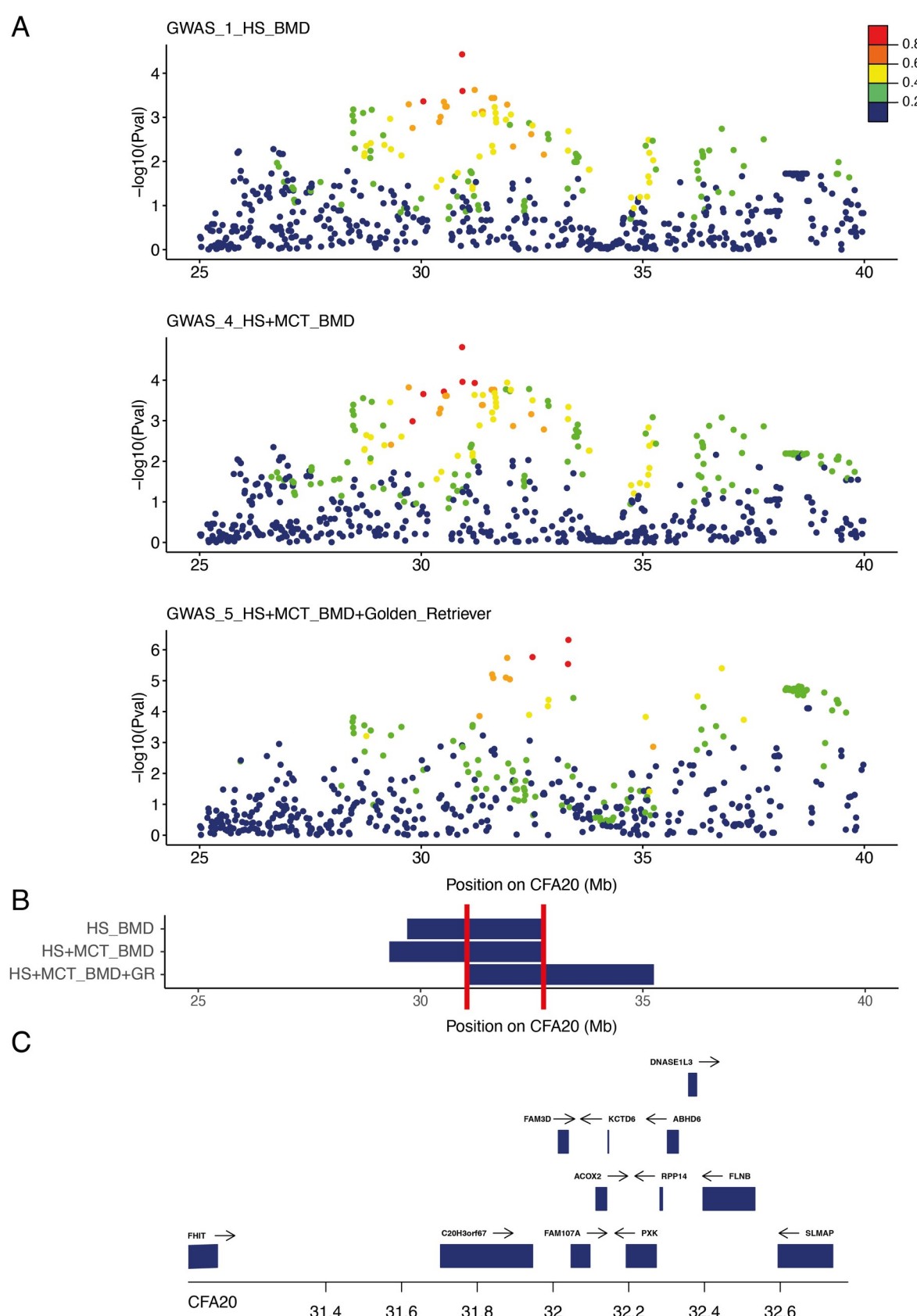

**Fig 3. Close up view of the CFA20 locus.** A) Manhattan plot of the CFA20 20–45 Mb region highlighting the best *p*-values obtained in the three genome-wide association study (GWAS): Bernese Mountain dogs (BMD) GWAS for histiocytic sarcoma (HS) with 172 cases vs. 128 controls (GWAS_1_HS_BMD); BMD GWAS for HS and mast cell tumor with 216 cases vs. 128 controls (GWAS_4_HS +MCT_BMD); meta-analysis combining BMD GWAS for HS and mast cell tumor with European golden retriever (69 cases vs. 74 controls) from Arendt et al. [15] (GWAS_5_HS+MCT_BMD+golden_retriever). The $R^2$ in cases from top single nucleotide variation (SNVs) show the linkage disequilibrium (LD) structure. B) Regions delimited by SNVs in LD with the best GWAS SNVs ($R^2 > 0.6$) in cases; minimal region between the three GWAS (CFA20:31036863–32778949) is delimited by red lines. C) Close up view of the genes (with available symbols) located in this minimal region (31–33 Mb) of CFA20.

The addition of 60 Rottweilers (37 cases and 23 controls) to GWAS_7_HS_BMD+-FCR_with_imputed_SNV formed a final dataset of 532,053 SNVs (GWAS_8_HS_BMD+FCR +Rottweiler_with_imputed_SNV), and led to the identification of 1,505 SNVs that were significantly associated with HS (Table 2, Fig 4E and 4F). This GWAS confirmed the involvement of the CFA2, CFA5, CFA11, and CFA14 loci in HS predisposition. SNVs in LD with significant SNVs in cases ($R^2 \geq 0.6$) allowed us to identify large regions spanning several Mb (up to 23 Mb for CFA11; Table 2). The analysis of these SNVs within the three breeds allowed us to reduce the CFA11 locus region to 38.4–41.4 Mb (Fig 4G).

These analyses identified SNVs that were significantly associated with HS risk and were shared between the three predisposed breeds on at least chromosomes 2, 5, 11, and 14. To

**Table 2. Significant loci identified by genome-wide association studies (GWAS) after imputation on high-density single nucleotide variation (SNV) array.** Number of associated SNVs with the best SNV and the corresponding corrected *p*-value are presented for each locus and each GWAS.

| Chromosome | | GWAS | | |
|---|---|---|---|---|
| | | BMD | BMD+FCR | BMD+FCR+Rott |
| 2 | Number of associated SNV | 1 | 4 | 7 |
| | Localisation of the best SNV | 29716535 | 29716535 | 29716535 |
| | | $p_{corrected} = 3.24 \times 10^{-4}$ | $p_{corrected} = 6.02 \times 10^{-5}$ | $p_{corrected} = 3.58 \times 10^{-5}$ |
| | Region delimited by significant associated SNVs | 29716535 | 29653137–29978776 | 29507029–34223001 |
| | Region delimited by significant associated SNVs and SNVs in LD in cases (R2≥0.6) | 29154373–30121047 | 29154373–30121047 | 29385904–35075340 |
| 5 | Number of associated SNV | 292 | 320 | 322 |
| | Localisation of the best SNV | 30496048 | 33823740 | 33823740 |
| | | $p_{corrected} = 8.22 \times 10^{-6}$ | $p_{corrected} = 2.52 \times 10^{-6}$ | $p_{corrected} = 2.4 \times 10^{-6}$ |
| | Region delimited by significant associated SNVs | 25628485–34513401 | 25522718–34477045 | 25566642–34477045 |
| | Region delimited by significant associated SNVs and SNVs in LD in cases (R2≥0.6) | 25402068–37781406 | 25402068–34513401 | 25517580–34513401 |
| 11 | Number of associated SNV | 1145 | 984 | 930 |
| | Localisation of the best SNV | 41215628 | 41252822 | 41252822 |
| | | $p_{corrected} = 1.46 \times 10^{-13}$ | $p_{corrected} = 1.49 \times 10^{-13}$ | $p_{corrected} = 2.02 \times 10^{-14}$ |
| | Region delimited by significant associated SNVs | 29934486–52418087 | 29978631–52418087 | 29978631–52418087 |
| | Region delimited by significant associated SNVs and SNVs in LD in cases (R2≥0.6) | 29047449–52471659 | 29315057–52471659 | 29341449–52418087 |
| 14 | Number of associated SNV | 292 | 250 | 246 |
| | Localisation of the best SNVs | 6567456 | 6567456 | 6566022 |
| | | $p_{corrected} = 4.05 \times 10^{-6}$ | $p_{corrected} = 1.37 \times 10^{-6}$ | $p_{corrected} = 1.09 \times 10^{-6}$ |
| | | 10231328 | 10665001 | 11021670 |
| | | $p_{corrected} = 9.19 \times 10^{-6}$ | $p_{corrected} = 2.78 \times 10^{-6}$ | $p_{corrected} = 1.52 \times 10^{-6}$ |
| | Region delimited by significant associated SNVs | 561549–11111293 | 561549–11111293 | 561549–11111293 |
| | Region delimited by significant associated SNVs and SNVs in LD in cases (R2≥0.6) | 475090–11638599 | 475090–11379670 | 475090–11379670 |

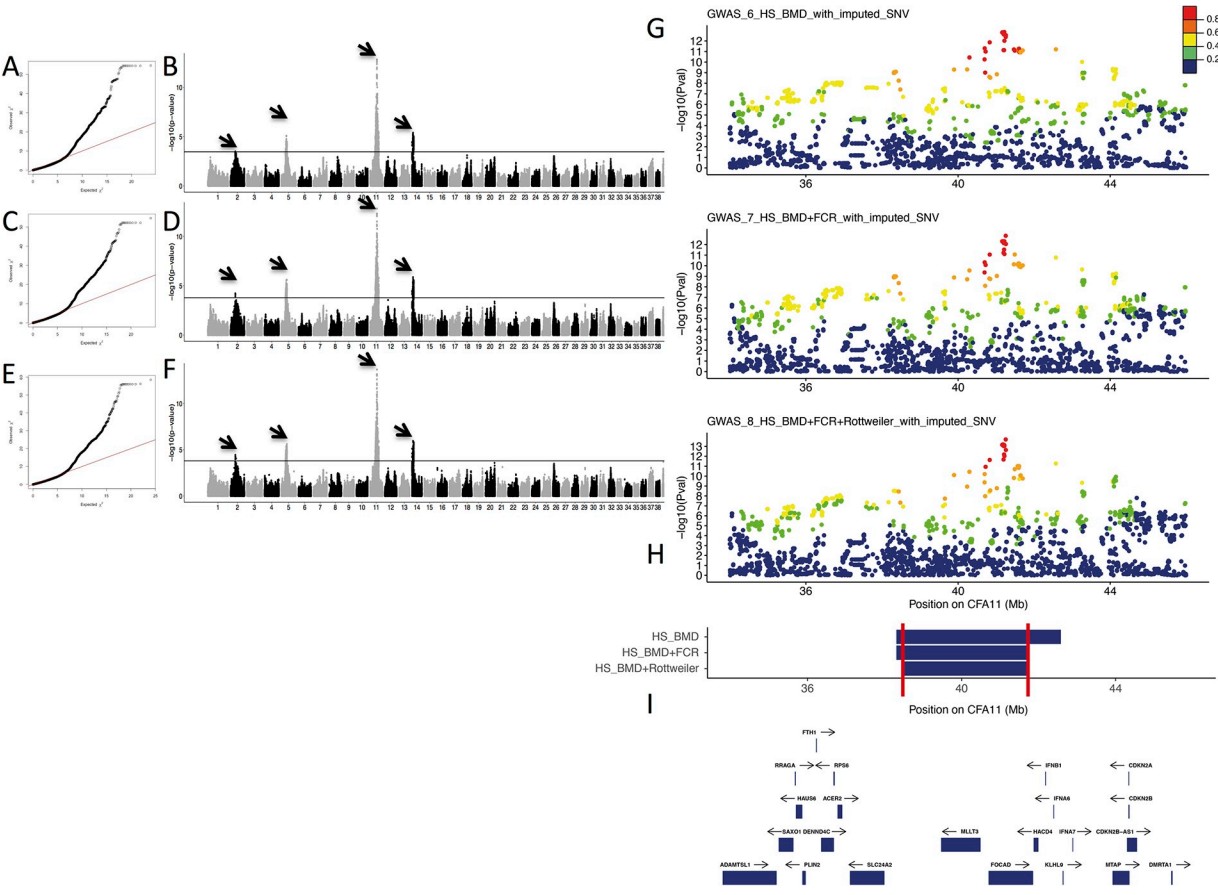

**Fig 4. Genome-wide association studies (GWAS) of Bernese mountain dogs (BMD) and other predisposed breeds on histiocytic sarcoma (HS) with the imputation of single nucleotide variations (SNV) on a higher density SNV array.** A–B) BMD GWAS results based on 403 cases and 347 controls (GWAS_6_HS_BMD_with_imputed_SNV). A) Quantile-quantile plot displaying a genomic inflation $\lambda$ of 1.000023, indicating no residual inflation. B) Manhattan plot displaying the statistical results from the GWAS. This analysis shows four loci (arrows) on chromosome 2 (best SNV at CFA2:29716535, $p_{corrected} = 3.25 \times 10^{-4}$), chromosome 5 (best SNV at Chr5:30496048, $p_{corrected} = 8.22 \times 10^{-6}$), chromosome 11 (best SNV at Chr11:41215628, $p_{corrected} = 1.45 \times 10^{-13}$), and chromosome 14 (CFA14:6567456, $p_{corrected} = 4.04 \times 10^{-6}$). C–D. GWAS results for HS combining BMDs (403 cases vs. 347 controls) and flat-coated retrievers (FCRs; 13 cases vs. 15 controls; GWAS_7_HS_BMD +FCR_with_imputed_SNV). C) Quantile-quantile plot displaying a genomic inflation $\lambda$ of 1.000018, indicating no residual inflation. D) Manhattan plot displaying the statistical results from the GWAS. This analysis shows four loci (arrows) on chromosome 2 (best SNV at CFA2:29716535, $p_{corrected} = 6.02 \times 10^{-5}$), chromosome 5 (best SNV at CFA5:33823740, $p_{corrected} = 2.52 \times 10^{-6}$), chromosome 11 (best SNV at CFA11:41252822, $p_{corrected} = 1.49 \times 10^{-13}$), and chromosome 14 (CFA14:6567456, $p_{corrected} = 1.37 \times 10^{-6}$). E–F) GWAS results for HS combining BMDs (403 cases vs. 347 controls), FCRs (13 cases vs. 15 controls), and Rottweilers (37 cases vs. 23 controls; GWAS_8_HS_BMD+FCR +Rottweiler_with_imputed_SNV). E) Quantile-quantile plot displaying a genomic inflation $\lambda$ of 1.000013, indicating no residual inflation. F) Manhattan plot displaying the statistical results from the GWAS. This analysis shows four loci (arrows) on chromosome 2 (best SNV at CFA2:29716535, $p_{corrected} = 3.58 \times 10^{-5}$), chromosome 5 (best SNV at CFA5:33823740, $p_{corrected} = 2.4 \times 10^{-6}$), chromosome 11 (best SNV at CFA11:41252822, $p_{corrected} = 2.04 \times 10^{-14}$), and chromosome 14 (best SNV at CFA14:6566022, $p_{corrected} = 1.09 \times 10^{-6}$). G) Close up view of the CFA11 locus highlighting the best $p$-values obtained in the three GWAS: BMDs GWAS (GWAS_6_HS_BMD_with_imputed_SNV), BMDs plus FCRs GWAS (GWAS_7_HS_BMD+FCR_with_imputed_SNV), and BMDs plus Rottweilers and FCRs GWAS (GWAS_8_HS_BMD+FCR +Rottweiler_with_imputed_SNV). $R^2$ in cases from top SNV is depicted to show the linkage disequilibrium (LD) structure. H) Regions delimited by SNVs in LD with the best GWAS SNVs ($R^2 > 0.6$) in cases, minimal region between the three GWAS (CFA11: 38435917– 41701130) is delimited by red lines. I) Close up view of the genes (with available symbols) located in this minimal region (38–42 Mb).

determine the proportion of HS risk that could be explained by these loci, we performed a restricted maximum likelihood (REML) analysis using GCTA software [40]. All chromosomes together could explain at least 61.8% of the phenotype ($p$-value $\leq 4.93 \times 10^{-29}$; S1 Table). SNVs of the CFA11 locus could explain over 10.3% ($p$-value $\leq 3.5 \times 10^{-19}$) of the HS phenotype; while, SNVs of the CFA14 locus explained a similar part of the phenotype, and the CFA5 could explain only 4.8–6.7% of the phenotype.

## 2.4. Haplotype analyses of HS loci identified risk haplotypes shared between breeds

To identify risk haplotypes tagged by the best SNVs and shared between HS cases, we determined the haplotype blocks including the best SNVs (Cf Materials and Methods), in each breed.

On CFA11, we identified a haplotype block containing the best CFA11 SNV (41,252,822) that was more frequent in HS cases in BMD and Rottweiler breeds than in the controls (0.79 vs. 0.55 and 0.77 vs. 0.54 in BMD and Rottweiler, respectively), and significantly linked to the risk of developing HS (odds ratio = 3.03, $p$-value = $7.03 \times 10^{-23}$ for BMDs; odds ratio = 2.61, $p$-value = 0.0175 for Rottweilers; Table 3). This block was frequently present in the FCR breed (53.8% and 56.6% in FCR cases and controls, respectively), and 75% of FCRs carried at least one copy of this risk haplotype. However, while the number of FCRs in the GWAS remained low, this haplotype did not appear to be enriched in HS cases (odds ratio = 0.89, $p$-value = 0.83). We identified a second independent CFA11 HS locus between 44 Mb and 45 Mb in the GWAS of the three breeds (GWAS_8_HS_BMD+FCR+Rottweiler_with_imputed_SNV; Fig 4G), which had already been identified in previous studies [26]. The best SNV in this region (CFA11:44,150,645) was not in LD with the best SNV of CFA11 (CFA11:41,252,822; $R^2$ of the three breeds = 0.41), indicating that there were at least two independent peaks on CFA11 involved in HS predisposition. Haplotype analysis of the CFA11:44 Mb region indicated a common risk haplotype enriched in the cases of the three breeds (0.66 vs. 0.44), and was significantly associated with the risk of developing HS (odds ratio = 2.45, $p$-value = $3.2 \times 10^{-19}$; Table 3).

For CFA5, the GWAS analysis indicated that a large locus (25–35 Mb) overlapped the two CFA5 lymphoma loci, as previously identified by Tonomura et al. (29 Mb and 33 Mb) [16]. Similarly, the haplotype analysis of the best SNVs indicated that the common risk haplotype was delimited by 18 SNVs (33,839,057–34,234,461) in the three breeds. This haplotype was significantly enriched in BMD and FCR cases, and was common in Rottweilers (69% and 63% in cases and controls, respectively; Table 3). In the three breeds, this risk haplotype was associated with a significantly increased risk of developing HS (odds ratio = 2.05, $p$-value = $6.58 \times 10^{-12}$).

For CFA14, the third main HS risk locus, GWAS analysis revealed a large region with significant SNVs spanning from 0.4 to 11.3 Mb. The associated SNVs in this region were clustered in at least two peaks located 4.4 Mb apart (Table 2). The top SNVs of these two peaks were located at 6,566,022 and 11,021,670 with $p_{corrected}$ = $1.09 \times 10^{-6}$ and $1.52 \times 10^{-6}$, respectively, in the GWAS including the three predisposed breeds (GWAS_8_HS_BMD+FCR+-Rottweiler_with_imputed_SNV). With an $R^2$ of 0.11 between these two SNVs across the three breeds, the data suggested that these peaks were independent. When considering the best SNV of CFA11 (CFA11:41,252,822), the two best SNVs of CFA14 were located at 10,665,001 ($p_{corrected}$ = $1.14 \times 10^{-7}$) and 11,021,670 ($p_{corrected}$ = $1.16 \times 10^{-7}$). Haplotype analysis of the region showed that the same risk haplotype was enriched in BMD and Rottweiler cases but not in FCR cases. Surprisingly, another haplotype, 5′-GCCAACGTATAAGT-3′, was enriched in the controls of the three breeds and was significantly associated with a decreased risk of developing HS in these predisposed breeds (odds ratio = 0.34, $p$-value = $9.08 \times 10^{-11}$). These results suggested that the CFA14 locus contains a protective allele, which is shared by the three predisposed breeds (Table 3).

Overall, this imputation with high density of SNVs allowed the identification of shared risk loci between the three HS predisposed breeds. We identified common risk or protective haplotypes shared by the predisposed breeds on major loci localized on CFA11, CFA14, and CFA5.

## 2.5. HS risk results from cumulative risk haplotypes

In these three breeds, cases had risk alleles on these three chromosomes, especially in BMD and Rottweilers, for which most cases had at least five risk copies (Table 4). In this cohort, no

**Table 3. Association analysis of haplotypes of CFA11, CFA5, and CFA14 loci with the phenotype in predisposed breeds.** The CFA11:41 Mb haplotype was determined by the genotype of the following SNVs: 41144469, 41161357, 41161441*, 41163558, 41166847, 41176819*, 41185205, 41192676, 41196587*, 41200012, 41204074, 41215628, 41217026*, 41218376, and 41252822. The CFA11:44 Mb haplotype was determined by the genotype of the following SNVs: 43263273, 43272660, 43290639, 43317981, 43690075*, 44070751, 44097553*, 44098601*, 44099662, 44121253, 44126921, 44129025, 44135605, 44150645*, 44152497, 44364615, 44366678, and 44367106*. The CFA5:33 Mb haplotype was determined by the genotype of the following SNVs: 33650581, 33788841*, 33801240*, 33807500*, 33823740*, 33839057, 33872506*, 33879286, 33883533, 33883991, 33897576, 33900059, 33906211, 33916864, 34213158, 34217528, 34220300, 34225552, 34225906, 34227643, 34231901*, 34232173, 34234461, 34239783*, 34240989, 34246822, and 34321500. The CFA14:11 Mb haplotype was determined by the genotype of the following SNVs: 11021670, 11023871, 11026224, 11028349, 11030215, 11033259, 11041572, 11067136, 11070626, 11077346, 11083217, 11092391, 11094795*, and 11108670. At risk haplotype in the breeds are represented in bold. CI: confidence interval (Woolf Method). * SNVs from the 173K SNV Canine HD.

| | | Haplotype frequencies in the 3 breeds | | | Odds Ratio | | |
| --- | --- | --- | --- | --- | --- | --- | --- |
| | | haplotype | affected | unaffected | Odds Ratio | confidence interval (Woolf Method) | pval |
| Locus: CFA11:41Mb | BMD | **ATTTAAAAAGCCC{A/G}T** | 0,79 | 0,55 | 3.03 | [2.42–3.79] | 7.03x10-23 |
| | | {A/G}GCCCGGGGAAGTGC | 0,21 | 0,45 | 0.32 | [0.26–0.4] | 2.56x10-23 |
| | | others | 0,00 | 0,00 | 1.73 | [0.32–9.47 | 0.69 |
| | Rottweiler | **ATTTAAAAAGCCCAT** | 0,76 | 0,54 | 2.61 | [1.19–5.73] | 0.0175 |
| | | AGCCCGGGGAAGTGC | 0,22 | 0,41 | 0.39 | [0.17–0.87] | 0.024 |
| | | others | 0,03 | 0,04 | 0.61 | [0.08–4.49] | 0.63 |
| | FCR | GTTTAAAAAGCCCAT | 0,54 | 0,57 | 0.89 | [0.31–2.56] | 0.83 |
| | | **GGCCCGGGGAAGTGT** | 0,38 | 0,2 | 2.5 | [0.76–8.25] | 0.12 |
| | | {A/G}GCCCGGGGAAGTAC | 0,08 | 0,23 | 0.27 | [0.05–1.44] | 0.15 |
| Locus: CFA11:44Mb | BMD | **ATGATAGGAACGGCAACT** | 0,65 | 0,44 | 2.41 | [1.96–2.97] | 7.27x10-17 |
| | | TCAGTTATTGTAATCGTC | 0,12 | 0,25 | 0.41 | [0.31–0.54] | 4.91x10-11 |
| | | TCAGTTATTGCGGTCGTC | 0,09 | 0,12 | 0.68 | [0.49–0.95] | 0.026 |
| | | TCAGCAATTGCGGTCGTC | 0,09 | 0,11 | 0.81 | [0.58–1.13] | 0.23 |
| | | others | 0,05 | 0,08 | 0.62 | [0.4–0.95] | 0.03 |
| | Rottweiler | **ATGATAGGAACGGCAG{T/C}C** | 0,82 | 0,57 | 3.61 | [1.57–8.33] | 0.003 |
| | | ATGACTATTGTAATCGTC | 0,07 | 0,17 | 0.34 | [0.1–1.11] | 0.078 |
| | | ATGATAATAGCGGCAACT | 0,05 | 0,15 | 0.32 | [0.09–1.16] | 0.1 |
| | | others | 0,05 | 0,11 | 0.47 | [0.12–1.85] | 0.30 |
| | FCR | **ATGATAGGAACGGCAA{T/C}T** | 0,35 | 0,17 | 2.65 | [0.76–9.29] | 0.137 |
| | | TCAGCAGGAACGGCAATC | 0,54 | 0,70 | 0.5 | [0.17–1.5] | 0.21 |
| | | others | 0,12 | 0,13 | 0.85 | [0.17–4.2] | 1 |
| | | | 1,00 | 1,00 | | | |
| | all breeds | **ATGATAGGAACGGCA---** | 0,66 | 0,44 | 2.45 | [2.01–2.99] | 3.2x10-19 |
| | | others | 0,34 | 0,56 | 0.41 | [0.34–0.5] | |
| Locus: CFA5:33Mb | BMD | **CTTTTCACACAAGTGTCCCGGTAGATT** | 0,75 | 0,59 | 2.05 | [1.65–2.55] | 1.01x10-10 |
| | | ACACCCGGGTTGACAGATTAACGACCT | 0,15 | 0,25 | 0.53 | [0.41–0.69] | 1.87x10-6 |
| | | ATTTCTGGGTTGACAGATTAACGACCT | 0,08 | 0,13 | 0.59 | [0.42–0.82] | 0.0016 |
| | | others | 0,02 | 0,03 | 0.7 | [0.37–1.32] | 0.69 |
| | Rottweiler | **CCACCCACACAAGTGTCCCGGTA{A/G}ATT** | 0,69 | 0,63 | 1.3 | [0.6–2.82] | 0.55 |
| | | CCACCCACACAAGTGTCCCGATAGATT | 0,12 | 0,17 | 0.66 | [0.23–1.85] | 0.66 |
| | | others | 0,19 | 0,20 | 0.96 | [0.38–2.44] | 0.96 |
| | FCR | **ACTTTCACACAAGTGTCCCGGTAGA{T/C}C** | 0,69 | 0,33 | 4.5 | [1.46–13.89] | 0.015 |
| | | CCTTCTACACAAGTAGACCGGTAACCT | 0,04 | 0,23 | 0.13 | [0.01–1.14] | 0.056 |
| | | CTTTCTGGGTTGACGTATTGACAGATT | 0,12 | 0,20 | 0.52 | [0.12–2.33] | 0.48 |
| | | others | 0,15 | 0,23 | 0.6 | [0.15–2.34] | 0.51 |
| | all breeds | **------CACACAAGTGTCCCGGTA-------** | 0,75 | 0,59 | 2.05 | [1.67–2.52] | 6.58X10-12 |
| | | others | 0,25 | 0,41 | 0.49 | [0.4–0.6] | |

(*Continued*)

**Table 3.** (Continued)

| | | Haplotype frequencies in the 3 breeds | | | Odds Ratio | | |
| --- | --- | --- | --- | --- | --- | --- | --- |
| | | haplotype | affected | unaffected | Odds Ratio | confidence interval (Woolf Method) | pval |
| Locus: CFA14:11Mb | BMD | **ATAGGAACCCGCGT** | 0,74 | 0,61 | 1.79 | [1.44–2.23] | 1.86x10-7 |
| | | ACCAACGCACGCAT | 0,17 | 0,19 | 0.85 | [0.65–1.1] | 0.25 |
| | | GCCAACGTATAAGT | 0,05 | 0,15 | 0.34 | [0.23–0.49] | 2.65x10-9 |
| | | ATAGGAACCCGCAT | 0,04 | 0,04 | 0.8 | [0.48–1.34] | 0.43 |
| | | others | 0,01 | 0,01 | 0.64 | [0.22–1.85] | 0.43 |
| | Rottweiler | **ATAGGAACCCGCGT** | 0,57 | 0,43 | 1.77 | [0.85–3.69] | 0.14 |
| | | ATAGGAACCCGCAT | 0,35 | 0,41 | 0.77 | [0.36–1.64] | 0.56 |
| | | GCCAACGTATAAGT | 0,07 | 0,11 | 0.59 | [0.16–2.16] | 0.59 |
| | | others | 0,01 | 0,04 | 0.2 | [0.02–1.98] | 0.15 |
| | FCR | **ACCAACGCACGCAC** | 0,46 | 0,20 | 3.43 | [1.05–11.17] | 0.036 |
| | | GCCAACGTATAAGT | 0,23 | 0,53 | 0.26 | [0.08–0.83] | 0.02 |
| | | ATAGGAACCCGCAC | 0,23 | 0,23 | 0.99 | [0.29–3.44] | 0.98 |
| | | others | 0,08 | 0,03 | 2.42 | [0.21–28.34] | 0.59 |
| | all breeds | GCCAACGTATAAGT | 0,06 | 0,16 | 0.34 | [0.24–0.47] | 9.08x10-11 |
| | | others | 0,94 | 0,84 | 2.92 | [2.09–4.08] | |

FCR controls had over three risk haplotypes. In the three breeds, the cumulative risk alleles on the three main loci (CFA11, CFA5, and CFA14) strongly impacted the probability of developing HS: carrying 5/6 risk alleles is associated with HS with an odds ratio of 5.27 ($p$-value = $1.52 \times 10^{-30}$). Stepwise model selections of the 1,505 significant SNVs identified by GWAS_8_HS_BMD+FCR+Rottweiler_with_imputed_SNV were performed to create generalized risk-models for HS (S2 Table). Both the Stepwise Forward model selection or the Stepwise

**Table 4. Association of CFA5, CFA11, and CFA14 with the phenotype in predisposed breeds.** The number of risk alleles was determined with the genotype of the following SNVs: CFA5:30496048, CFA11:41252822, and CFA14:11021670. CI: confidence interval (Woolf Method).

| | frequencies of risk alleles in the 3 breeds | | | Odds Ratio | | |
| --- | --- | --- | --- | --- | --- | --- |
| | number of risk alleles | affected | unaffected | Odds Ratio | confidence interval (Woolf Method) | pval |
| BMD | $\geq$5 risk alleles | 0,72 | 0,31 | 5.67 | [4.14–7.76] | 1.44x10-29 |
| | 4 risk alleles | 0,23 | 0,4 | 0.46 | [0.34–0.63] | 1.779x10-6 |
| | $\leq$ 3 risk alleles | 0,05 | 0,29 | 0.12 | [0.07–0.2] | 1.34x10—20 |
| Rottweiler | $\geq$5 risk alleles | 0,74 | 0,43 | 3.51 | [1.17–10.53] | 0.03 |
| | 4 risk alleles | 0,18 | 0,26 | 0.66 | [0.19–2.29] | 0.53 |
| | $\leq$3 risk alleles | 0,08 | 0,3 | 0.2 | [0.05–0.88] | 0.034 |
| FCR | 4 risk alleles | 0,38 | 0 | NA | NA | 0.013 |
| | 3 risk alleles | 0,54 | 0,53 | 1.02 | [0.23–4.52] | 1 |
| | $\leq$2risk alleles | 0,08 | 0,47 | 0.1 | [0.01–0.98] | 0.037 |
| all breeds | $\geq$5 risk alleles | 0,70 | 0,31 | 5.27 | [3.92–7.08] | 1.52X10-30 |
| | 4 risk alleles | 0,23 | 0,37 | 0.51 | [0.38–0.69] | 1.45x10-5 |
| | $\leq$3 risk alleles | 0,07 | 0,32 | 0.15 | [0.1–0.23] | 8.1X10-22 |

Forward and Backward model selection included markers of CFA2, CFA5, CFA11, and CFA14, with several markers spaced more than 4 Mb apart. The fact that some markers are separated by several Mbases on each chromosome, shows the cumulative effect of several independent risk haplotypes.

## 2.6. Capture and targeted sequencing of the three best HS candidate loci (CFA5, CFA11, and CFA14)

Since common haplotypes were detected in the three predisposed breeds on the three main loci (CFA5, CFA11, and CFA14), we sequenced these three regions to identify putative common variants. DNA samples from 16 dogs (10 BMDs, 4 Rottweilers, and 2 FCRs) from the three predisposed breeds, with a balanced distribution of risk and protective haplotypes, were selected for targeted sequencing. An average of 9,458 SNVs and 2,674 indels per sample were identified with a mean depth of 142 x per sample. These variants (SNVs and indels) were imputed on the remaining dog samples (453 HS cases and 385 controls from BMD, Rottweiler, and FCR breeds). A total of 2,608 significant variants (of which 1,886 were on CFA11) were identified while performing statistical analysis with the imputed genotypes. For CFA11, no coding variants were significantly associated with HS risk, and four of the top ten variants associated with HS predisposition were imputed genotypes and localized within 100 kb (S3 Table). The best-associated variant remained CFA11:41,252,822, which was already identified in the previous GWAS (see section 2.3.). CFA11:41,252,822 is localized in a non-coding transcript (CFRNASEQ_UC_00018829) that overlaps *CDKN2A* and CDKN2B antisense RNA (*CDKN2B-AS1*). Interestingly, looking at RNASeq data from Hoeppner et al [41], we found that CFRNASEQ_UC_00018829 was highly expressed in blood than in eight other tissues. Moreover, the human orthologous region of this SNV (chr9:21,996,622, hg38), close to the CpG island (chr9:21,994,103–21,995,911, hg38) and DNAse I hypersensitivity peak cluster (chr9:21,994,641–21,996,130, hg38), overlaps an enhancer (GH09J021996, chr9:21,996,543–21,996,791, hg38), which regulates *CDKN2A* and cyclin-dependent kinase 4 inhibitor B (*CDKN2B*). The fourth top variant was an indel that is localized in a non-coding transcript (CFRNASEQ_IGNC_00021613), 6,500 bp upstream of the *CDKN2A* transcript (S3 Table). Since it was previously shown that the expression of *CDKN2A* correlated with CFA11:41 Mb risk haplotype [26], we strongly suspected that these non-coding SNVs could deregulate the expression of *CDKN2A*.

To identify the best candidate variants in the secondary loci, a complementary association analysis was performed by including the genotypes of the best variant (CFA11:41,252,822) as a covariate (S3 Table). We identified 2,413 variants (of which 21 SNVs were on CFA2, 1,533 SNVs were on CFA5, and 859 SNVs were on CF14) with significant residual associations in the three breeds. Residual association was found for the CFA11:44–45 Mb locus with the best SNV CFA11:45,941,548 ($p$-value = 0.008), close to C9orf72, and not in LD with CFA11:41,252,822 ($R^2$ of the three breeds = 0.0899). This result confirmed the existence of two independent risk loci on CFA11 between 41–45 Mb.

For the CFA14 locus, seven of the top ten variants were imputed variants, and were localized between positions 10,665,001 and 11,042,153. This locus overlapped POT1 antisense RNA 1 (*POT1-AS1*) and protection of telomeres 1 (*POT1*).

In the CFA5 locus, nine of the top ten variants of CFA5 that were found to be associated with the risk of developing HS were imputed variants; they were localized between positions 30,483,338 and 30,496,048 that overlap the *SPNS3* gene (S3 Table and S4 Fig). Interestingly, the second (CFA5:30,489,203) and third (CFA5:30,489,217) variants were localized in a region containing DNA methylation marks, and one of them included the CFA5:30,489,217 variant.

Moreover, the nearby CFA5:30,489,203 variant created a CpG site. Thus, we hypothesized that these two close SNVs could be associated with allele-specific methylation in histiocytic cells. Bisulfite sequencing of HS cell lines confirmed that these two variants (CFA5:30,489,203 and CFA5:30,489,217) presented with specific alleles (CFA5:30,489,203-G and CFA5:30,489,217-C, respectively), thereby creating CpG sites with methylation in histiocytic cells (S3 Fig). This was also the case for the best HS GWAS SNV on CFA2 with the CFA2:29,716,535-G allele (S5 Fig). We hypothesized that these specific methylation alleles could be associated with modifications of regulation of neighboring genes expression.

## 2.7. Validation of major loci on independent cohorts

Since risk haplotypes were detected in the three predisposed breeds because of the imputed data, we genotyped variants of the three major loci on an independent cohort of BMD (186 cases and 176 controls) to validate the major role of these loci in HS (S4 Table). This analysis confirmed that the risk alleles of the top variants of CFA5, CFA11, and CFA14 (see section 2.4) significantly increased the risk of developing HS (odds ratios = 2.56–3.94, $p$-value = $2.4 \times 10^{-6}$–$6.93 \times 10^{-16}$). Moreover, when a BMD case from this cohort carried 5/6 risk alleles, it strongly impacted the probability of developing this cancer with an odds ratio of 9.71 ($p$-value = $3.44 \times 10^{-23}$).

Since a majority of BMDs succumb to cancer, mostly HS, lymphomas, or mast tumors [26,30,35], we suspected that carrying these risk alleles would impact the life span of BMDs. We thus analyzed the risk alleles on an independent cohort of 317 dogs (age <10 years and without pathological diagnosis of HS (see Materials and methods) and correlated longevity with the number of risk alleles. This analysis confirmed that, independently of the known clinical status, carrying 5/6 risk alleles significantly impacted the longevity at the BMD population level (median = 7.5 vs 9.17 years, $p$-value = 0.0015, log-rank test; S6 Fig).

## 3. Discussion

### 3.1. Dog breeds: unique models to detangle the genetic features of human cancers

Several genetic studies of canine cancers have shown that high-risk dog breeds can lead to the advancement in genetics of rare cancers in humans [6,8,25]. Indeed, a limited number of critical loci have been identified in canine cancers [6,15–17,26], wherein some loci are shared by several canine cancers, and are also well-known in human cancers. Moreover, somatic alterations identified to date in canine tumors through genome-wide approaches, are found in the same genes [25,42,43]. HS affects a few dog breeds with incredibly high frequencies (BMD, Rottweiler, retrievers), and these breeds appear to be a perfect example to study the genetics underlying such a strong HS predisposition in dogs.

Further, dissecting the genetic factors for rare cancers such as HS is challenging in humans. Hence, we proposed that dog models of HS could help in deciphering the genetic predisposition factors of this rare cancer. Previously, we had successfully identified a relevant locus on canine chromosome 11, and this locus is also well-known in canine osteosarcoma and several human cancers (human chr.9p21 locus) [17,44]. In this study, we presented the GWAS on a large cohort of several dog breeds affected by HS. We applied a multiple-breed approach to not only refine the previously identified locus on CFA11 but also identified additional loci on CFA2, CFA5, CFA14, and CFA20. Moreover, this study highlighted the fact that besides an initial association signal peak found in canine cancer GWAS, the independent risk haplotypes can be cumulative and shared by several dog breeds and in several cancers.

## 3.2. Genetic predisposition of canine cancers: cumulative risk haplotypes

In humans, the genetic architecture of cancer risk is usually described by a combination of rare variations in families with dominant inheritance patterns, and common variants with small-effect sizes in the population. Within-breed canine GWAS usually identifies fewer variants with stronger effects [18] because of breed structure and artificial selection of variants with strong effects, which is not necessarily the rule for cancer development. Indeed, because of the genetic drift of canine breeds resulting from the strong selection of the morphological criteria of sires and dams used for reproduction, deleterious alleles could be involuntarily selected and enriched in specific populations. Consequently, significant associations detected with cancer in dog GWAS could be due to cumulative risk alleles. In such conditions, it would be surprising if HS predisposition was only due to one risk haplotype, especially since at least 33 loci have been identified by previous GWAS for canine osteosarcoma predisposition [17]. Our results confirmed the main role of the CFA11 locus, along with the cumulative effects of at least two different risk haplotypes as suggested in our previous study [26]. Here, a multi-breed approach allowed to refine the main risk haplotype on CFA11 to a region of ~74 kb that was shared between BMDs and Rottweilers.

Moreover, this study allowed the identification of a strong candidate variant overlapping *CDKN2B-AS1* that regulates *CDKN2A*. Additional GWAS peaks were also identified on CFA2, CFA5, CFA11, CFA14, and CFA20. Some of these loci were shared between the three predisposed breeds, mostly between Rottweilers and BMDs, which was expected considering the close phylogenetic relationship between these two breeds [45]. FCR are a small number of dogs in France; thus, the number of FCR samples included in the study was low, and further GWAS will be needed to better decipher the shared predispositions between FCR and other breeds. Nevertheless, in these three predisposed breeds, the cumulative risk alleles on the three main loci (CFA11, CFA5, CFA14) strongly impacted the probability of developing this cancer with an odds ratio of 5.27, i.e., to be affected by HS when dogs carry 5/6 risk alleles. This study illustrated that the GWAS association detected between a cancer and a locus in dogs could hide the cumulative risk of several haplotypes. This study also confirmed previous findings in the golden retriever breed by Arendt et al. (2015) and Tonomura et al. (2015) who described that there were at least two independent risk haplotypes on the CFA20 locus for mast cell tumor and on the CFA5 locus for lymphoma, respectively [15,16].

## 3.3. Multi-cancer loci identified through a multi-breed approach

Additionally, we confirmed that some risk haplotypes were also involved in several cancers, as suggested by Tonomura et al., based on the association of the CFA5 locus with hemangiosarcoma and lymphoma [16]. Such pleiotropy at cancer risk loci has also been observed in human cancers, where one-third of the SNVs mapped to genomic loci are associated with multiple cancers [44]. Here, we confirmed that the multi-cancer effect of loci for CFA5, CFA11, and CFA20 influenced the risk of HS, lymphomas, osteosarcomas, and mast cell tumors in BMD or golden retrievers. However, further studies with higher SNV density in the golden retriever breed are needed to confirm whether the same predisposing risk alleles are shared with BMD. The *CDKN2A* locus was detected in Rottweilers and BMDs, and a neighboring region of the locus (CFA11:41.37 Mb; canFam3) was associated with osteosarcoma and was fixed in the Rottweiler population [17]. This shows that HS-affected Rottweilers were accumulating risk haplotypes for at least two cancers (osteosarcoma and HS) at this locus. Co-occurrence of two different risk haplotypes for HS and osteosarcoma across 200 kb also perfectly illustrates that a given locus can harbor multi-cancer risk because of different risk haplotypes as observed in humans. Indeed, in human GWAS, for some loci such as 8q24.21 (containing *MYC*), different risk SNVs are associated with different risk cancers, although they might ultimately converge

toward the same oncogenic mechanism [44]. In such a situation, using GWAS in a multi-breed strategy can help decipher risk haplotypes when several dog breeds share several predispositions.

## 3.4. Pleiotropic effect of loci

Cancers are multigenic diseases wherein cumulative alterations in key pathways are considered as hallmarks of cancer [46]. HS involves histiocytic immune cells, and is suspected to be at the crossroads of immune dysregulation and cancer predisposition in dogs. HS predisposed breeds, especially BMD, are also predisposed to reactive histiocytic diseases [47], and other immune or inflammatory diseases such as glomerulonephritis, aseptic meningitis, and inflammatory bowel disease (https://www. bmdca.org/health/diseases.php) [48,49]. While no causal relationship was found between inflammation and HS, inflammation is suspected to contribute to HS development [50–52]. Thus, it is not surprising that HS GWAS hits overlap not only candidate tumor suppressor genes (*TUSC1*; tumor suppressor candidate 1) or other well-known tumor suppressors involved in cell cycle (*CDKN2A*), genome stability (telomere protection: *POT1*, replication stress/DNA damage: *FHIT*) but also inflammation (*IL17rd*, *SPNS3*, *ARHGEF3*; S5 Table). The GWAS hits highlighted an enrichment of genes involved in cancer pathways (cell cycle, $p$-value = $9.28 \times 10^{-3}$; cellular senescence, $p$-value = 0.013; bladder cancer, $p$-value = 0.049; aging, $p$-value = 0.0024; signaling pathways regulating pluripotency of stem cells, $p$-value = 0.0066; TP53 network, $p$-value = 0.03) and lipid metabolism (regulation of lipid metabolism by peroxisome proliferator-activated receptor alpha [PPARalpha], $p$-value = 0.016; response to leptin, $p$-value = 0.0066). Expanding our search for HS association signals clearly showed overlaps with human GWAS signals (S5 Table). A number of these genes are not only known to be involved in the predisposition of several cancers (*CDKN2A*, *POT1*, *FHIT*) but are also associated with immune traits (monocyte, platelet, etc.), cholesterol, high density lipids/light density lipids, and allergens in humans. This suggests that the pleiotropic nature of these loci is not limited to cancer risk. This is apparent in humans; for instance, loci associated with N-glycosylation of human immunoglobulin G show pleiotropy with autoimmune diseases and hematological cancers [53]. Concomitantly to this work, a study by Labadie et al. (2020) confirmed the pleiotropic effect of these canine cancer loci by identifying a shared region for canine T zone lymphoma, mast cell tumors, and hypothyroidism in golden retriever; one of these loci on CFA14 that is involved in mast cell tumors and canine T zone lymphoma is ~800 kb downstream of *POT1* locus identified in this study [54].

## 3.5. Genetic predisposition of HS in three most at risk dog breeds (CFA2, CFA5, CFA11, CFA14, and CFA20)

In addition to the CFA11 major locus, this study identified other HS loci on CFA2, CFA5, CFA14, and CFA20. Imputation was done to perform GWAS on a larger cohort of cases and controls from the three predisposed breeds with a higher density of SNVs. To avoid incorrect genotypes and potentially unreliable results after imputation, we used parameters previously shown to give accurate imputation in dogs [39]. Both BMD and FCR breeds were included in the reference panel of the higher density SNVs, imputation was performed within the same breeds, and simulations showed a good concordance of the imputed genotypes (91.97–95.86%). However, while there is a close genetic relationship between BMDs and Rottweilers [45,55], the lack of Rottweilers in the reference panel may be a potential limitation. Nevertheless, the shared risk haplotypes identified in the three predisposed breeds had SNVs only from the Illumina 173K SNV Canine HD array; thus, none of them were imputed in the FCR or

Rottweiler cohorts (Table 3). Finally, we replicated these results on two independent cohorts of BMDs, and showed the reliability of the loci and risk alleles identified in this study, and their impact on the longevity of BMDs in the whole population.

The CFA5 locus is associated with HS and lymphoma risk in BMDs, and the top SNVs of this locus are located in an intron of the *SPNS3* gene. However, very little is known about this gene although its paralogous gene (*SPNS2*) is important in immunological development, and plays a critical role in inflammatory and autoimmune diseases, influences lymphocyte trafficking and lymphatic vessel network organization, and drives defective macrophage phagocytic functions [37,56]. Moreover, this region is associated in human GWAS with chemokine CLL2 [57]. Thus, *SPNS3* is a strong candidate gene that can explain the predisposition to HS and lymphoma.

Further, the CFA14 locus has already been suggested in our previous study [26]. Here, the best SNVs are located in the introns of *POT1-AS1* and *POT1*. *POT1* encodes a nuclear protein that is involved in telomere maintenance and in the predisposition and development of numerous cancers (S5 Table). This study identified a potential protective haplotype that was shared between the three predisposed breeds. The difference in age during onset of cases carrying zero or one copy, and the difference in age at death of controls carrying none, one, or two copies of this protective haplotype suggested that this shared protective haplotype was most probably involved in longevity, and thereby in the age of onset of HS (S7 Fig).

CFA20 locus is associated with HS and mast cell tumor risk in BMDs. The top SNV here is present in the intron of *FHIT*, a tumor suppressor involved in apoptosis and prevention of the epithelial-mesenchymal transition, and one of the earliest and most frequently altered genes in most human cancers [38], including predisposition in breast cancer [57]. Moreover, stable nuclear localization of FHIT is a special marker for histiocytes, suggesting another function of FHIT as a signaling molecule related to antiproliferation function [58].

### 3.6. Causal variants

Like previous studies on canine HS predisposition [26] and human cancers [44,59], capture and targeted sequencing of the three HS candidate loci did not allow the identification of potential coding variants and risk variants that are common in the predisposed breeds. While the genotype concordance of imputed SNVs with the Beagle software was reliable, imputation of SNVs identified by the capture could limit the detection of causal variants. However, as in humans, majority of the loci identified from cancer GWAS do not directly affect the amino acid sequence of the expressed protein, and thus elucidation of causal variants is challenging, since all closely linked variants that are in LD with the best GWAS SNV are relevant candidates [44]. In this study, the best SNVs were either in the intronic part of the candidate genes (*FHIT*, *SPNS3*, etc.), upstream of a candidate gene (*POT1*), or overlapped long non-coding RNAs near a strong candidate gene (*CDKN2A*, *POT1*). Finally, we showed that the best variants linked to *SPNS3* or *PFKFB3* belonged to CpG sites methylated in histiocytic cells. This strongly suggests that HS predisposing variants in dogs are non-coding variants with regulatory effects. Under these conditions, the local cumulative risk haplotypes could reflect complex regulatory interactions such as the *MYC* locus, for which recent Hi-C analysis of its genomic region has demonstrated a complicated regulatory mechanism, thereby implicating that various large intergenic non-coding RNAs may mediate effects at the risk loci [44]. Therefore, further functional studies are needed to identify the involvement of such variants in the regulation of candidate genes.

### 3.7. Conclusions

In conclusion, we presented the largest GWAS of HS in dog cohorts through a multi-breed approach, and confirmed the main role of the *CDKN2A* locus (CFA11/HSA9q21), and

identified four new loci (CFA2, CFA5, CFA14, and CFA20). We used multiple breeds and cancers to highlight the cumulative effect of different risk haplotypes behind each locus and their pleiotropic nature. Finally, while capture and targeted sequencing of specific loci did not lead to the identification of straightforward variants linked to cancer predisposition, our results pointed toward strong candidate genes and to the existence of regulatory variants in the non-coding regions and CpG islands linked to risk haplotypes, which lead to strong cancer predispositions in specific dog breeds.

## 4. Materials and methods

### 4.1 Ethics statement

The study was approved by the Centre National de la Recherche Scientifique (CNRS) ethical board, France (35-238-13).

### 4.2. Sample collection

Blood and tissue biopsy samples from cancer-affected and unaffected dogs were collected by a network of veterinarians through the Cani-DNA Biological Resource Center (http://dog-genetics.genouest.org), and DNA and RNA samples were extracted as previously described [10]. Samples were collected by veterinarians during the medical care of the dogs, with the informed consent of the owners. Blood and tissue samples were collected at a medical visit or during surgery, and then stored in tubes containing EDTA or RNAlater, respectively. We selected dogs based on their breeds i.e., BMD, Rottweiler, and FCR. For the cases, we included dogs with a pathology report confirming diagnosis of hematopoietic cancer (HS, mast cell tumor, or lymphoma), and for controls, we included dogs >10 years who were without cancer.

### 4.3. GWAS

DNA samples were genotyped on an Illumina 173K SNV Canine HD array at the Centre National de Génotypage (Evry, France) and on Affymetrix Axiom Canine Genotyping Array Set A and B 1.1M SNV array at Affymetrix, Inc. (Santa Clara, CA, USA; Table 1). SNV genotypes were filtered with pre-sample call rate >95%, per SNV call rate >95%, and minor allele frequency (MAF) = 0.01. Breed check was performed with a cluster tree and genetic matrix distances obtained from common SNVs of our data and in publicly available data [15,16]. Sex check was performed via the Plink 1.9 option "—*check-sex*"[60]. For GWAS including multiple breeds, we conducted this quality control protocol for each breed before merging the dataset. We used mixed linear model analyses by taking into account the population structure and kinship with R package "Eigenstrat–GenABEL" 1.8 [61] on R studio software (version 1.1.463; Vienna, Austria). *P*-values corrected for inflation factor λ were used, and we identified all SNVs with significant association exceeding 95% confidence intervals as defined empirically using 1,000 random phenotype permutations with the Eigenstrat–GenABEL 1.8 software [61].

For imputation, the Beagle software [62] version 4.1 was used to impute the Illumina 20K SNV and Illumina 170K SNV arrays to the Affymetrix 712K SNV array format as well as to input 21,614 variants identified from the capture and targeted sequencing of 16 dogs. Thereafter, 113 BMDs and 21 FCRs, representative of the BMD and FCR populations, were selected and genotyped on the Affymetrix 712K SNV array. These dogs were used as a reference panel for the imputation. SNVs for imputation were filtered for MAF > 0.05 and Hardy-Weinberg Equilibrium *p-value* > $1 \times 10^{-7}$, as previously described by Friedenberg and Meurs [39]. All default settings of the Beagle software were used except for the following options: *niterations = 50 window = 3000*.

## 4.4. Capture and sequencing of targeted canine GWAS loci

In total, four loci were captured and sequenced for 16 dogs selected according to their haplotype: one localized on CFA5 (29,805,467–34,459,320), two on CFA11 (41,148,019–41,237,204 and 43,520,875–46,778,525), and one on CFA14 (9,949,911–11,524,424).

Capture, sequencing, variant detection, and annotation were performed by IntegraGen S.A. (Evry, France). Genomic DNA was captured using Agilent in-solution enrichment methodology via the Agilent SureSelect Target Enrichment System kit (Agilent technology, Santa Clara, California, USA). The SureSelect Target Enrichment workflow is a solution-based system that uses ultralong 120-mer-biotinylated cRNA baits to capture regions of interest by enriching them out of a next-generation sequencing genomic fragment library. Library preparation and capture were followed by paired-end 75 base massive parallel sequencing on an Illumina HiSeq 2000 sequencer [63].

A custom-made SureSelect oligonucleotide probe library was designed to capture the loci of interest according to Agilent's recommendations with 1× and 2× tiling densities using the eArray web-based probe design tool (https://earray.chem.agilent.com/earray). A total of 57,205 RNA probes were synthesized by Agilent Technologies, Santa Clara, CA, USA. Sequence capture, enrichment, and elution were performed according to the manufacturer's instructions and protocols (SureSelect, Agilent) without any modification, except library preparation, which was performed instead with NEBNext Ultra kit (New England Biolabs). For the library preparation, 600 ng genomic DNA was fragmented by sonication and purified to yield fragments of 150–200 bp. Paired-end adaptor oligonucleotides from the NEB kit were ligated on repaired A-tailed fragments, then purified and enriched by eight polymerase chain reaction (PCR) cycles. Next, 1200 ng of these purified libraries were hybridized to the SureSelect oligo probe capture library for 72 h. After hybridization, washing, and elution, the eluted fractions were PCR-amplified with nine cycles, purified, and quantified by quantitative PCR (qPCR). Based on this quantification, an equimolar pool was acquired and quantified again by qPCR. Finally, the pool was sequenced on an Illumina HiSeq 2000 platform as paired-end 75 bp reads. Image analysis and base determination were performed using the Illumina RTA software version 1.12.4.2 with default settings.

Bioinformatics analyses of sequencing data were based on the Illumina pipeline (CASAVA 1.8.2). CASAVA performs an alignment of a sequencing run to a reference genome (canFam3), calls SNVs based on allele calls and read depth, and detects variants (SNVs and indels). The alignment algorithm used was ELANDv2 (Maloney alignment and multi-seed reducing artifact mismatches). Only the positions included in the bait coordinates were conserved. Genetic variation annotation was performed using IntegraGen in-house pipelines. It consisted of gene annotations (RefSeq), detection of known polymorphisms (dbSNP), and variant annotation (exonic, intronic, silent, nonsense, etc.).

## 4.5. Predictive modeling

For model selection, we extracted the data only for the 1,505 significant SNVs identified by GWAS_8_HS_BMD+FCR+Rottweiler_with_imputed_SNV, and recoded them as 0 for AA genotype, 1 for AB genotype, and 2 for BB genotype. For the first model selection method, we used a Stepwise Forward selection based on a 0.05 alpha inclusion and exclusion threshold inspired from Zapata et al [20]. Briefly, the selection process began with a model with no terms. Independent variables were sequentially added based on their lowest p-values in the generalized model. Before adding the next term, the selection method would remove any variables that became non-significant after the inclusion of the previous term. The selection process was terminated when no more terms could be added or removed from the model. For the

second model selection, we used a stepwise both forward and backward model selection using the R library MASS [64] to select the best predictive model. To reduce the number of SNV to test, a first selection was performed by adding in the generalized model all significant SNVs identified in Table 2 and removing any variables that became non-significant. The Stepwise both forward and backward model selection was performed on the 44 remaining significant SNVs.

## 4.6. REML analysis

Estimation of the phenotypic variance for HS based on genetic variance was performed by REML analysis using GCTA [40] with default settings. In our analyses, variance of a genetic factor was determined by the genotypes of SNVs on all autosomes and within the associated regions on chromosomes 5, 11, and 14. Log-likelihood ratio tests were performed with no estimated prevalence, since the prevalence of HS is unknown in Rottweiler and FCR breeds, and is 0.25 for BMDs.

## 4.7. Haplotype analyses

Minimal risk haplotypes for different breeds were identified on the associated loci. First, variants in strong LD ($R^2 > 0.8$) with the top SNVs were identified in each breed using PLINK 1.9 [60] LD clumping, and used as an input for haplotype phasing in each breed with fastPHASE 1.4. [65]. Risk haplotypes enriched in cases were identified based on the top SNV genotype. Starting from the top SNV localization, and then moving both up- and downstream, we identified the SNV positions where the risk haplotype was broken by a recombination event (i.e., two alternative alleles were present on both the risk and non-risk haplotypes). This was done separately for each breed, and thereafter, the minimal shared risk haplotype across breeds was defined.

## 4.8. Gene-set enrichment analysis

Approved symbols of closest genes to the strongest significant signal per chromosome and per GWAS were analyzed with the online tool Webegestalt [66] to identify overrepresentation of pathways in the identified loci of the GWAS.

## 4.9. Methylation analyses

Methylation analyses were performed on 12 HS cell lines, including one commercial cell line (DH82; American Tissue Type Culture, CRL-10389; RRID:CVCL_2018) and 11 cell lines developed from HS-affected fresh dog tissue samples. These eleven cell lines are available on request (S6 Table). The cells were cultivated in complete Roswell Park Memorial Institute Medium (RPMI) medium containing RPMI 1640 GlutaMAX supplemented medium (Gibco Life Technologies) with 10% fetal bovine serum (HyClone, GE Healthcare, Life Sciences, Logan, UT) and 0.025% primocin (InvivoGen, Toulouse, France) at 37°C in a humidified 5% $CO_2$ incubator. All cell lines were tested for mycoplasma using the MycoAlert Plus kit (Lonza, Rockland, ME) by following the manufacturer's protocol, and were found to be mycoplasma-free cells. SNVs of CFA5 and CFA2 were sequenced by Sanger sequencing as previously described [10] with the following primers: CFA2_29716535-F: 5′-GGTGTACTTTCGGGTCC AAC-3′, CFA2_29716535-R: 5′-CCCTGTCATTCGATGTCCTT-3′, CFA5_30489203–3048 9217_F: 5′-CCTGAGTGAGTGGAATGAGGA-3′, CFA5_30489203–30489217_R, 5′-CTTC CTGCGACCTGCTGT-3′ in absence and/or (CFA2_29716242–29716795_FM: 5′-TAGGTGT TGGGTTTATATTGTTAGG-3′, CFA2_29716242–29716795_RM: 5′-CTTCCTGCGACCT

GCTGT-3′, CFA5_30488738–30489339_FM: 5′-TAGGTGTTGGGTTTATATTGTTAGG-3′, CFA5_30488738–30489339_RM: AAACCTATTCTCTTTTTCTAATTCACTTTA) in the presence of bisulfite conversion (EZ DNA Methylation-Gold Kit, Ozyme, St Cyr–l'école, France).

## 4.10. Genotyping for validation on independent cohorts

Genotyping of the SNVs of Chr5_30488886, Chr11_41252822, and Chr14_11021670 was performed by targeted sequencing using Ion AmpliSeq technology with Ion GeneStudio S5 Prime System (Life Technology, ThermoFisher Scientific, Courtaboeuf, France). Probes were designed using the AmpliSeq design service, and Torrent Suite Software (v5.12) was used for sequencing data processing. Torrent Mapping Alignment Program (TMAP) software was used to perform read processing and mapping on the loaded genome (canFam3) using default parameters. Variant Caller Plugin was used for variant calling, and two independent cohorts were selected from the BMD samples collected between 2012–2020. The first independent cohort for genetic analyses was made of 186 cases and 176 controls. The second independent cohort for survival analyses comprised 317 dogs without a pathological report of HS and younger than 10 years (thus, independent of the first cohort). Survival probability was estimated using the Kaplan–Meier method, and the differences in longevity according to the number of risk alleles were tested using the log-rank test via the "survival" package of R [67].

## Supporting information

**S1 Fig. Pedigrees of a Bernese mountain dog family** showing the co-segregation of lymphoma (blue) and mast cell tumor (green) with histiocytic sarcoma (black). (TIFF)

**S2 Fig. Genome-wide association studies (GWAS) on histiocytic sarcoma (HS) and lymphoma.** A–B. Bernese mountain dog (BMD) GWAS results for HS with 172 cases and 128 controls (GWAS_1_HS_BMD). A) Quantile-quantile plot displaying a genomic inflation λ of 1.000005, indicating no residual inflation. B) Manhattan plot displaying the statistical results from the GWAS. This analysis shows two loci (arrows) on chromosome 11 (CFA11:41161441, $p_{corrected} = 3.11 \times 10^{-7}$) and chromosome 20 (CFA20:30922308, $p_{corrected} = 3.73 \times 10^{-5}$). C–D. BMD GWAS results for HS and lymphoma with 252 cases vs. 128 controls (GWAS_2_HS+lymphoma_BMD). C) Quantile-quantile plot displaying a genomic inflation λ of 1.000005, indicating no residual inflation. D) Manhattan plot displaying the statistical results from the GWAS. This analysis shows two loci (arrows) on chromosomes 11 (CFA11:41161441, $p_{corrected} = 1.5 \times 10^{-6}$) and 5 (CFA5:30496048, $p_{corrected} = 5.88 \times 10^{-6}$). E–F. Meta-analysis combining the BMD GWAS for HS and lymphoma (252 cases vs. 128 controls) and the golden retrievers GWAS for lymphoma (41 cases vs. 172 controls) from Tonomura et al. [16] (GWAS_3_HS+lymphoma_BMD+golden_retriever). E) Quantile-quantile plot displaying a genomic inflation λ of 1, indicating no residual inflation. F) Manhattan plot displaying the statistical results from the GWAS. This analysis shows the locus on chromosome 5 (CFA5:32824053, $p_{corrected} = 2.2 \times 10^{-7}$). (TIFF)

**S3 Fig. Genome-wide association studies (GWAS) on histiocytic sarcoma (HS) and mast cell tumor.** A–B. Bernese mountain dog (BMD) GWAS results for HS with 172 cases and 128 controls (GWAS_1_HS_BMD). A) Quantile-quantile plot displaying a genomic inflation λ of 1.000005, indicating no residual inflation. B) Manhattan plot displaying the statistical results from the GWAS. This analysis shows two loci (arrows) on chromosome 11 (CFA11:41161441,

$p_{\text{corrected}} = 3.11 \times 10^{-7}$) and chromosome 20 (CFA20:30922308, $p_{\text{corrected}} = 3.73 \times 10^{-5}$). C–D. BMD GWAS results for HS and mast cell tumor with 216 cases vs. 128 controls (GWAS_4_HS +MCT_BMD). C) Quantile-quantile plot displaying a genomic inflation λ of 1.000005, indicating no residual inflation. D) Manhattan plot displaying the statistical results from the GWAS. This analysis shows two loci (arrows) on chromosomes 11 (CFA11:41161441, $p_{\text{corrected}} = 6.94 \times 10^{-7}$) and 20 (CFA20:30922308, $p_{\text{corrected}} = 1.53 \times 10^{-5}$). E–F. Meta-analysis combining the BMD GWAS for HS and mast cell tumor (216 cases vs. 128 controls) and European golden retriever GWAS for mast cell tumor (69 cases vs. 74 controls) from Arendt et al. [15] (GWAS_5_HS+MCT_BMD+golden_retriever). E) Quantile-quantile plot displaying a genomic inflation λ of 1, indicating no residual inflation. F) Manhattan plot displaying the statistical results from the GWAS. This analysis shows the locus on chromosome 20 (CFA20:33321282, $p_{\text{corrected}} = 4.79 \times 10^{-7}$).
(TIFF)

**S4 Fig. Identification of DNA methylation sites at the single nucleotide variations (SNVs) on the CFA5 locus included in CpG islands.** The UCSC track of chr5:30,489,183–30,489,230 with methylation track (Dog-MDCK-Meth) and Sanger sequencing performed on histiocytic sarcoma cell lines are represented. Sequencing of homozygous and heterozygous histiocytic sarcoma cell lines in the absence and presence of bisulfite treatment showed that the two SNVs present with allele-specific methylation in histiocytic cells.
(TIFF)

**S5 Fig. Identification of DNA methylation sites at the single nucleotide variations (SNVs) on the CFA2 locus included in CpG islands.** The UCSC track of chr2:29,716,519–29,716,550 with methylation tracks (Dog-MDCK-Meth, Dog-R3-Sperm-Meth) and Sanger sequencing performed on histiocytic sarcoma cell lines are represented. Sequencing of homozygous and heterozygous histiocytic sarcoma cell lines in the absence and presence of bisulfite treatment showed that the SNV presents with allele-specific methylation in histiocytic cell lines.
(TIFF)

**S6 Fig. Impact of risk alleles on the life span of Bernese mountain dogs (BMD).** Kaplan–Meier estimates of BMD longevity and the corresponding hazard ratio is represented according to the number of risk alleles ($n \leq 4$ or $n \geq 5$). Survival mean and survival median are 8.4 and 9.5 years, respectively, for BMDs with $\leq 4$ risk alleles; whereas, the survival mean and survival median are 7.54 and 7.92 years, respectively, for BMDs with $\geq 5$ risk alleles.
(TIFF)

**S7 Fig. Age of onset or death of Bernese mountain dog (BMD) histiocytic sarcoma (HS) cases and controls according to the number of copies of protective CFA14 haplotype.** Cases with zero copies, n = 183, mean age = 6.22 years. Cases with one copy, n = 24, mean age = 7.23 years. Cases with two copies, n = 1, mean age = 6.9 years. Controls with zero copies, n = 132, mean age = 11.21 years. Controls with one copy, n = 50, mean age = 11.27 years. Controls with two copies, n = 5, mean age = 11.97 years. One-sided Wilcoxon rank sum test was conducted.
(TIFF)

**S1 Table. Variance explained by chromosomes 5, 11, 14, or all autosomes, as estimated by restricted maximum likelihood (REML) analysis.**
(XLSX)

**S2 Table. Risk modeling of histiocytic sarcoma (HS).**
(XLSX)

**S3 Table. Top 10 variants identified in chromosomes 11, 14, and 5 loci after imputation of captured variants on 455 histiocytic sarcoma (HS) cases and 408 controls from Bernese mountain dog (BMD), Rottweiler, and flat-coated retriever (FCR) breeds.** For chromosomes 14 and 5, the association analyses were performed by taking into account the information of the best SNV of chromosome 11 (CFA11:41252822).
(XLSX)

**S4 Table. Association analysis of CFA11, CFA5, and CFA14 loci with the phenotype in an independent validation cohort of Bernese mountain dog (BMD; 186 cases and 176 controls). At-risk alleles are represented in bold.** CI: confidence interval (Woolf method).
(XLSX)

**S5 Table. List of candidate genes identified by genome-wide association studies (GWAS) performed on several dog breeds and several cancers.** Indicated genes correspond to the closest genes with the best single nucleotide variation (SNV) per GWAS experiment. The involvement of these genes in human cancer or inflammation is summarized according to the NIH gene database (https://www.ncbi.nlm.nih.gov/gene). Their association with specific traits in human GWAS is summarized according to the PheGenI database (https://www.ncbi.nlm.nih.gov/gap/phegeni).
(XLSX)

**S6 Table. List of cell lines developed from HS-affected fresh dog tissue samples.** The breed, the tumoral tissue and mutation status for *PTPN11*, *KRAS* and *BRAF* are indicated.
(XLSX)

## Acknowledgments

We thank the veterinarians who provided anatomopathological diagnoses, especially Olivier Albaric and Laetitia Dorso (Laboniris, Oniris, Ecole Nationale Vétérinaire de Nantes, France), Marie-Odile SEMIN (LAPV, Amboise, France), Caroline Laprie (Vet-Histo, Marseille, France), Marie Lagadic (Idexx Alfort, France), and Frédérique Degorce-Rubiale (LAPVSO, Toulouse, France). We also thank the veterinarians for providing us with clinical data and samples as well as the dog owners, breeders, and breed clubs, especially the French club AFBS, European clubs, American club BMDCA, IWG International Working Group (for Bernese mountain dogs), and US Berner-Garde foundation, and especially Pat Long for her dedicated trust and follow-up of our study. We also warmly thank Clotilde de Brito, Laetitia Lagoutte, Annabelle Garand, Anne Sophie Guillory, Melanie Rault, and Ronan Ulve (IGDR, Rennes, France) for their technical support and helpful discussions, Stéphane Dréano (IGDR, Rennes, France) for Sanger sequencing and Biogenouest bioinformatics platform. Finally, we thank Guillaume Queney and Anne Thomas from Antagene (Animal Genetics Company, La Tour de Salvagny, France) for providing samples and genetic sample characterization, as well as Eleonore Thierry, for her follow-up of BMD dogs in the frame of her DVM at Antagene.

## Author Contributions

**Conceptualization:** Benoît Hédan, Patrick Devauchelle, Jérôme Abadie, Catherine André.

**Data curation:** Benoît Hédan, Maud Rimbault.

**Formal analysis:** Amaury Vaysse, Nadine Botherel.

**Funding acquisition:** Benoît Hédan, Catherine André.

**Investigation:** Benoît Hédan, Maud Rimbault, Thomas Derrien.

**Methodology:** Benoît Hédan, Édouard Cadieu, Thomas Derrien.

**Project administration:** Benoît Hédan.

**Supervision:** Benoît Hédan.

**Validation:** Édouard Cadieu, Caroline Dufaure de Citres.

**Writing – original draft:** Benoît Hédan, Maud Rimbault, Pascale Quignon, Thomas Derrien, Catherine André.

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
