## [Decision Letter · Decision Letter 0]

8 Sep 2020

Dear Drs. André and Hédan,

Thank you very much for submitting your Research Article entitled 'The tree that hides the forest: identification of common predisposing loci in several hematopoietic cancers and several dog breeds.' to PLOS Genetics. Your manuscript was fully evaluated at the editorial level and by independent peer reviewers. The reviewers appreciated the attention to an important problem, but raised some substantial concerns about the current manuscript. Based on the reviews, we will not be able to accept this version of the manuscript, but we would be willing to review again a much-revised version. We cannot, of course, promise publication at that time.

In particular, reviewers were unable to understand the design or logical order underlying the overall narrative. They indicated there may be meaningful discovery here, but suggested a major re-write could be most appropriate if their critical technical and statistical concerns were rigorously resolved. Based on the reviews, you could reduce the scope to focus on the strongest findings or could clarify the full study. Statistical significance, possibility of imputation problems and lack of specificity in referring to genetic markers, alleles and haplotypes must be addressed satisfactorily. The issue of lack of replication or other supporting evidence was raised. While this standard for publication of human genetics may not be required for animals, this limitation must be addressed if it were relevant (with suggestions of necessary follow-on studies). Please confirm the data will be deposited in a public archive with free access. Reviewer suggestions that are desirable include consideration of effect sizes and heritability.

If you decide to revise the manuscript for further consideration at PLOS Genetics, please aim to resubmit within the next 60 days, unless it will take extra time to address the concerns of the reviewers, in which case we would appreciate an expected resubmission date by email to plosgenetics@plos.org.

[LINK]

We are sorry that we cannot be more positive about your manuscript at this stage. Please do not hesitate to contact us if you have any concerns or questions.

Yours sincerely,

Carlos E Alvarez

Guest Editor

PLOS Genetics

Gregory Barsh

Editor-in-Chief

PLOS Genetics

Reviewer's Responses to Questions

**Comments to the Authors:**

Reviewer #1: I consider focusing on germline for loci in these canine cancers and ultimately variants conferring susceptibility quite important and adding together different breeds a strength. The datasets are now becoming quite large and able to find interesting results. Q-Q plots indicate that proper corrections for populations have occurred and I have confidence that the SNP genotype data are treated properly. Here is what I understand: loci for different cancers can be shared within a breed and loci for the same cancer can be shared across breeds; a known locus has been further defined and additional loci detected; and the underlying variants are likely not in coding regions of genes.

Major comments.

But I confess to having a lot of difficulty in following the author’s presentation. After struggling I ended up questioning the logical order the authors use to tell their story. The progression of 7 GWA with adding breeds and cancer types using just the 173K SNP data was really a load to try to understand and the presentation in general did not help. Having to look for Tables 1 – 3, while the Figures were inserted in the text did not help.

I suggest that the authors consider starting off right at the clearest analysis and the ones with the most strength; i.e., the large population sizes and imputed data from several lo density arrays on HS alone (GWA 8) that leads to Figure 6. And follow it with the HS GWA that adds in FCR and Rottweilers (GWA 9 and 10). With that strongest and clearest result in hand it seems that some of the figures and analyses with lower density SNPs leading up to it may not be necessary. And then go on to the stories of combining the HS with MCT and Lymphoma.

The pedigree of Figure 2 showing cosegregation of cancers may not tell the whole story. Conclusions on a single pedigree can be influenced by ascertainment bias. And, how many of the dogs have more than one form of cancer? Can the numbers of dogs with each or both conditions across the entire population be provided instead? Anyway I suggest moving Figure 2 to Supplemental.

Often relies on “top SNPs” and their p values, but converting this into understanding by the reader that these SNPs are tagging haplotypes and that haplotype distributions will be useful is not consistent. For example, is the CFA5 locus necessarily the strongest or most influential locus, or are the marker allele frequencies and its haplotype structure in the populations making it more easily discovered by GWA?

Can the authors better provide and explain the haplotype information, numbers and frequencies between cases and controls and how this information resulted in simple regions defined in the figures? Better presenting the complexity of genomic landscape would be very helpful. Better explain the content and results from Tables 2 and 3 in the text. And if there is a limitation to number of Figures/Tables allowed in the main text, shift more of the allotment to the haplotype and loci tables that are now in the Supplement.

Please bring out the story around the haplotypes where at most the frequency of a cancer associated haplotype is twice that in the cases than in the controls. Anyway, the discussion of the haplotypes in the text were not enough for me.

I believe this wealth of information can be used to better quantitate the heritability and genetic architecture of HS and perhaps the other 2 cancers as well. The SNP genotype data from the cases and controls could be used to estimate narrow sense heritability (Yang, J., Lee, S.H., Goddard, M.E. and Visscher, P.M. (2011) GCTA: a tool for genome‐wide complex trait analysis. Am. J. Hum. Genet. 88, 76‐82.)

Other comments:

Intro on page 6. Transitioning between somatic tissue studies and mutations found there and GWA studies for underlying susceptibility loci in a very long paragraph is not smooth. Perhaps the use of the term “genetic bases” is not specific enough.

Paragraph 2.1 When stating that the 3 loci are also found / or overlapping in GWA of other cancers can the authors find a way to better demonstrate to us that the regions truly overlap? As it stands they do not give the sequence coordinates routinely and just state a gene name or the chromosome number.

Paragraphs are missnumbered in Table 1.

It is not clear to me how the authors can claim 3 independent peaks on CFA5 for HS and Lymphoma in the combined breeds (Figure 3).

Reviewer #2: The authors present a Genome-Wide Association Study that evaluates the risk of HS in a cohort of BMDs that was collected through a collaborative network of veterinarians. Samples collected were genotyped using the Illumina 173k and Affymetrix 1.1m SNP arrays that were further imputed for uniformity. GWAS of this data was sequentially complemented with the addition of genotype data from previously collected data for other cancers including Lymphoma and Mast Cell Tumor Cancer. These data sets were used to discover common SNP variants associated to combined cancer risk. Additional GWAS were performed where several breeds were added to refine the analysis. An evaluation of the imputed data was also included along with a sequencing effort to detect functional variants and a CNV assessment of risk of the chr14 loci.

-Overall assessment based on the PLOS GENETICS criteria for publication with a numeric scoring by the reviewer (1-5, five being best):

-Originality (2): The presented paper is very similar to others already published such as Karlsson et al 2013 and Tonomura et al 2015 (cited by the study). It does incorporate some new elements but overall they do not detect highly significant findings that are relevant to the field.

-Importance (2): The reviewer believes that the dog is a very important emerging model for studying cancer. However, this field has been stagnant for some years where discoveries are not followed up. The novel loci discovered in this study a likely going to follow that trend. This is a sad reality of the field.

-Interest to researchers (4): this type of approach is likely to be relevant to researchers in the field and is likely to be cited. Dog papers are very popular.

-Rigor (4): The reviewer believes the data was rigorously evaluated as the methods are quite standard in the field. However, the paper requires further clarification of methodological details and a more through evaluation of the assumptions used (i.e. imputation bias of minor allele frequency SNPs)

-Evidence (2): The study adds a couple candidates to the list of cancer risk genes but does not provide additional evaluation and critique of these new findings. In comparisons to similar papers such as Karlsson and Tonomura's, this paper looks weak.

Major concerns:

The paper requires a deep revision to correct grammar and typos. In addition, the paper is written in a very verbose and unnatural language that makes some arguments hard to understand. The reviewer strongly recommends a native English speaker to review and improve the syntax in the paper. There are too many instances to enumerate but some examples are offered at the end.

The variants detected in this study are in large part the same as those already discovered for other cancers (such as osteosarcoma by Karson el al in 2013, cited in the paper). This main concerns is even clearly stated by the authors in section 3.4 "A number of these genes are not only already known to be involved in the predisposition of several cancers (CDKN2A, POT1, FHIT,…) but they are also associated with immune traits (monocyte, platelet…) or with cholesterol, HDL/LDL or allergens traits in Human". Curiously, this phenomenon is attributed to a pleiotropic effect (see section 3.3); however, also it is discussed in the same section a few lines after, how there are two variants within a 200kb region where one is fixed in the Rottweiler. This argument is counter intuitive because if there are two variants, how can it be pleiotropy?

Although the GWAS implicated other novel loci associations, no additional biological argument supported by any other type of technology different from GWAS was provided to support their inclusion. No independent validation cohorts were used.

It is undisclosed where the samples were collected from. Accounting for samples collected by the study and those added from previously published studies become complicated as they are progressively added into the argument. There are no clear descriptions of the samples in the methods and results that help account for samples included in the study. i.e. The reviewer believes the FCR samples were collected by the researchers but is not completely sure.

The imputation effort made in the study is concerning even when it has some caveats on his favor. It appears to the reviewer that a 20k genotype dataset was imputed and expanded to a 712k as described in section 4.2. To the reviewer’s opinion, the magnitude of this imputation effort is extreme and very risky. However, to the researcher’s favor, they did this within a breed that are likely to be fixed over large portions of the genome. The reviewer believes it is impossible to determine how appropriate is this approach within the context of dog genomics and the study. In contrast and considering the Minor allele frequency exclusion of 0.01, in humans such threshold would still leave a significant risk of biased SNPs for low frequency SNPs (see Figure 2 in Johnson et al 2013 PMID: 23334152, attached). Although the reviewer considers this section to be plausible, it would be appreciated to have more details to evaluate imputation bias in this specific dog cancer and compared it to a more generalized context.

Even when the sequencing effort was made to detect functional variants, there was no significant findings obtained from it, this is clearly stated in section 3.6. This dramatically limits the impact and relevance of the effort.

For the figures, the corrected P-values mentioned in the text do not correspond to what is represented in the figures in the respective vertical axis. i.e. Figure 1 chr5 loci is 6.36E-5 while in the figure is barely above the E-4. This occur in all figures.

For figure 2, besides being a complete chaos of vertical and horizontal mating, the color coding is not legible when printed in a high quality color laser printer. These two issues are likely to make this figure completely useless for the reader.

Some arguments in the paper that are concerning to the reviewer:

In the introduction "and models are strongly needed to better understand this dramatic cancer" --- This a weird statement and an example of the odd writing style used in the paper. What is a dramatic cancer?

In the introduction "Furthermore, despite a strong/important heritability of HS in BMD, the awareness of this devastating cancer and attempt to selection against HS since 20 years, breeders have not succeeded to reduce the prevalence of this cancer" --- What evidence does the authors have that suggest there has been any effort to reduce prevalence?

In section 2.3 "since it is never found in other 231 Swiss dog breeds (18 Appenzeller Sennenhund, 8 Entlebucher Mountain dog and 205 greater Swiss dog, data not shown)..." --- This sentence contains an obvious mistake that makes the sentence very confusing 231 Swiss dog breeds, more like 231 dogs belonging to Swiss breeds. Also, what is the relevance of the Swiss breeds to the BMD and Rottweiler argument when the Rottweiler is not a Swiss breed.

In section 2.5 "When performing the statistical analysis with the imputed genotypes, no coding variant significantly associated with HS risk could be found and six of the ten top variants associated with HS predisposition are imputed genotypes and are localized within 100 kb on CFA11" --- Could this be due to the extreme imputation effort made in the study that is previously described as a concern by the reviewer?

In section 3.2 " the cumulative risk alleles on the 3 main loci (CFA11, 5, 14) strongly impact the probability to develop this cancer with an Odds ratio of 5.41 to be affected by HS when dogs carry 5/6 risk alleles" --- This is an important finding to the reviewer that is not described or discussed.

In section 3.6 "With multiple breeds and cancers approach, we highlight the cumulative effect of different risk haplotypes behind each locus and their pleiotropic nature" --- the reviewer is of the opinion that to make this statement it would have been necessary a modeling approach such as the one presented by Zapata et al 2019 where cumulative effects are expressly evaluated and collinearity is considered.

Reviewer #3: Comments to the author:

This study aims to identify loci associated with histiocytic sarcoma in the three dog breeds most at-risk for this cancer. To increase GWAS statistical power, the authors:

a) increase sample size by including phenotypes for several closely related cancers (like lymphoma and mast cell tumor), making use of previously published data,

b) increase sample size by combining cases and controls from several breeds known to have a predisposed risk (like golden retriever and Rottweiler)

c) increase the number of SNPs to test by using several different arrays (ranging from a 22k custom capture array to the 1.1M Affymetrix array) and imputing between these.

Significantly associated loci on 8 different chromosomes are identified – some also found in previous studies – and the authors follow up on several of these by investigating haplotypes, using targeted sequence capture and performing methylation analyses. The authors find several risk (or protective) haplotypes that are mostly shared between breeds and some even between the different cancer types, and which are regulatory in nature. This is a thorough and interesting research study, which propels the field of canine complex disease genetics.

The manuscript is generally written well and easy to follow. But there are many minor English grammatical errors.

Main comments:

1. Phenotyping – no details at all are provided on how cases and controls were selected. What criteria were used (apart from breed)? Was there a minimum age for controls?

2. Quality control – what other filters did you use on your genotyping data? Only MAF of 1% is mentioned. What about SNVs that failed Hardy-Weinberg? Or SNVs with high missingness? Did you do a sex check and a breed check (like a PCA) to make sure there were no sample mixups? What was the final SNV number left for analysis after these filters? More details are required.

3. Permutations – I’m a little concerned that the thresholds were all set based on only 500 permutations. 10,000 is a much more reasonable and usual number, but at least 1,000 permutations should be used. Also, more details about how these permutations were run is needed – program used, etc. And how was the threshold actually calculated? Some of the “significant” P-values are very low…certainly much lower than the Bonferroni correction (although realizing this is overly conservative).

4. There are no imputation accuracy metrics given or tests run. Or even previous papers showing canine imputation accuracy cited. How do we know the imputation worked well and provides reliable results?

Minor comments: (Note: there are no page numbers or line numbers so that makes this difficult)

1. I’m not a fan of the first part of the title

2. Write out the gene names the first time they are used

3. Introduction, 2nd paragraph – “clinical presentation, histopathology and issue of this canine cancer…” – not sure what is meant by “issue”, remove.

4. Introduction, 2nd paragraph – “attempt to selection against HS since 20 years, breeders have not succeeded to reduce the prevalence” should be “attempt to select against HS for 20 years, breeders have not succeeded in reducing the prevalence”

5. Introduction, 3rd paragraph – “showed that they are also associated to other cancers in predisposed breeds” should be “with other cancers”

6. I think fig. 2 should be supplemental – it doesn’t show your results and is more like a side-note

7. Fig. 4 – addition of 44 cases with MCT to the HS data. But the number of controls decreases from 154 to 135 – why?

8. Section 2.3 last paragraph – “specific to BMD and Rottweilers HS cases since it is never found in other 231 Swiss dog breeds” should be “specific to BMD and Rottweiler HS cases since it is never found in a further 231 dogs of other Swiss breeds”

9. Section 2.3 last paragraph, last line – should be “close” not “closed”

10. Fig 5 & 6 - remove question marks in boxes next to A, B, C, etc

11. Section 2.4, 1st paragraph – 134 dogs were genotyped on Affymetrix array – more detail about the dogs selected – what breeds? Add to methods.

12. Section 2.4, 4th paragraph “no common risk haplotype can been identified with higher SNV…” should be “can be identified”

13. Section 2.4, last three words should be “Supplementary Table 5”.

14. Section 2.5, 1st paragraph – §3.4 doesn’t seem relevant to this sentence.

15. Section 2.5 – either use commas or spaces to separate the millions and thousands in bp coordinates but not both

16. Section 2.5 – the top ten variants are shown but how many significant SNVs were actually identified for each of these GWAS?

17. Section 2.5, 2nd paragraph – where is the data from the GWAS using the CFA11 best SNV as a covariate? Add the top ten results to Supp. Table 6.

18. Discussion – small “h” on human

19. Discussion, 3.1, 2nd sentence – switch the words “common” and “loci”

20. Discussion, 3.1, 3rd sentence needs rewriting. Maybe: “Interestingly, somatic alterations identified to date in canine tumors, through genome-wide approaches, are found in the same genes”

21. Discussion, 3.2 title – remove the “of” after cumulative

22. Discussion, 3.2, 2nd sentence needs rewriting. Maybe: “Within-breed canine GWAS usually identifies fewer variants with stronger…”

23. Discussion, 3.2, 3rd sentence – replace “beauty” with “morphological”

24. Discussion, 3.2, middle of the paragraph – need a fullstop after “Rottweilers” and then on the same line remove the repeated words “a strong”

25. Discussion, 3.2, last line – add in years for Arendt et al and Tonomura et al

26. Discussion, 3.4 – please put P-values into scientific notation and remove the … in the lists in brackets

27. Discussion, 3.4, last line – start needs rewriting. Maybe: “Concomitantly to this work, a study by Labadie et al confirmed the pleiotropic effect of these canine cancer loci by identifying a shared region for canine T zone lymphoma,…”

28. Discussion, 3.5, 2nd sentence – use “The CFA5 locus…” 3rd sentence – use “Very little is known…”

29. Discussion, 3.5, 1st paragraph, reference needed for sentence ending in “associated in human GWAS with the chemokine CCL2.”

30. Discussion, 3.5, 2nd paragraph – use “difference in age” instead of “difference of the age”

31. Discussion, 3.5, 3rd paragraph – use “The CFA20 locus..” Reference needed for sentence ending in “involved in human predisposition of breast cancer.”

32. Supplementary figure 5 caption – change all occurrences of “copy” to “copies” after either zero or two, i.e., should be “zero copies” and “two copies”. Also, second occurrence of “Cases with one copy” should be “Cases with two copies” and second occurrence of “Controls with one copy” should be “Controls with two copies”.

33. Methods – give the full name for abbreviations, like Cani-DNA BRC, CNRS, CNG

34. Methods, 4.1 – change the Ulve et al citation to the corresponding reference number

35. Methods, 4.1, 3rd sentence – should be “veterinarians” and no s on “medical care”

36. Methods, 4.2 – what numbers of dogs from what breeds were genotyped on the different arrays? Details missing.

37. Methods, 4.2 – GenAbel 1.8 – is this an R package? If so, you should state this.

38. Methods, 4.3, 3rd paragraph – small q on qPCR

39. Methods, 4.3, last sentence should read: “It consists of gene annotations (RefSeq), and detection…”

40. Table 1 – paragraph numbers are wrong

41. Supplementary table 6 – please explain the last column, Pc1df

42. Supplementary table 7 – it would help if you added the chromosomes (and approx. bp) for these genes

**Have all data underlying the figures and results presented in the manuscript been provided?**

Reviewer #1: Yes

Reviewer #2: **No: **Just a note on privacy. The authors offered to make the data available in their private research team page. Although this page is public, it is still possible for the authors to track people who access it from their IPs. Within the close community of dog genomics researchers, it becomes very easy to determine the identity of possible competitors accessing the data and thus provide an unfair advantage to the authors.

Reviewer #3: **No: **But the authors state that the genotypes will be available upon publication

PLOS authors have the option to publish the peer review history of their article (what does this mean?). If published, this will include your full peer review and any attached files.

Reviewer #1: No

Reviewer #2: No

Reviewer #3: No

---

## [Decision Letter · Decision Letter 1]

3 Feb 2021

Dear Dr Hedan,

We are pleased to inform you that your manuscript entitled "Identification of common predisposing loci to hematopoietic cancers in four dog breeds" has been editorially accepted for publication in PLOS Genetics. Congratulations!

Yours sincerely,

Carlos Alvarez

Guest Editor

PLOS Genetics

Gregory Barsh

Editor-in-Chief

PLOS Genetics

Comments from the reviewers (if applicable):

Please note that Fig. 4.I has all genes in upper case, except for "cdkn2A". The text also refers to CDKN2B and "CDKN2A-AS", but they are absent from this figure. CDKN2B-AS1 is well known, but I am not aware of "CDKN2A-AS"; please check the symbol of the AS gene you refer to. Let us know promptly if you decide to make changes to text or figure. -- Carlos Alvarez, Guest Editor

Reviewer's Responses to Questions

**Comments to the Authors:**

Reviewer #1: The clarity and flow has greatly improved. The authors have addressed each of my concerns.

One problem is that the text in several of the Tables is messed up, perhaps during conversion to a pdf?

Reviewer #2: First of all, the authors dramatically improved the language and overall written quality of the manuscript. The language used in current revision sounds natural and clear.

The addition of clear subject numbers at the beginning of each argument and the addition of table 1 are quite helpful for following and keeping track of the exact subjects used for each study section. The authors addressed the concerns and requests for clarification for other missing or confusing arguments.

Table 4 is a great addition to the manuscript along with the addition of the additional text in section 2.5. In addition, the addition of the expanded and rearranged discussion talking about the pleiotropic effects in section 3.4 and genetic predisposition in section 3.5 is much appreciated.

Although the reviewer has still some concern on the impact of such an extended imputation effort as a generalized approach in dog genomics, the rationale provided by the authors and the specific circumstances of this present study are reasonable to address such concern. The reviewer believes that these large approaches must be evaluated for appropriateness on a case by case basis where the specifics of the cohort, the reference panel and an “a posteriori” re-evaluation of the appropriateness of the imputation effort based on the amount of hits relying only in imputed SNPs. As mentioned in the first revision, the reviewer believes this approach is plausible and for this study, the new additions, corrections and additional context provided by the authors satisfy his concern for him to recommend the manuscript for publication.

Reviewer #3: The authors have made substantial improvements to the manuscript. I thank them for taking the time to thoroughly address all my concerns.

The tables (especially table 2) are hard to read because of all the question marks in boxes, but it sounds like this is due to a pdf conversion error, and will be sorted out by editorial staff.

**Have all data underlying the figures and results presented in the manuscript been provided?**

Reviewer #1: Yes

Reviewer #2: Yes

Reviewer #3: Yes

PLOS authors have the option to publish the peer review history of their article (what does this mean?). If published, this will include your full peer review and any attached files.

Reviewer #1: No

Reviewer #2: No

Reviewer #3: No

**Data Deposition**

http://datadryad.org/submit?journalID=pgenetics&manu=PGENETICS-D-20-01160R1

**Press Queries**

---

## [Editor Report · Acceptance letter]

26 Feb 2021

PGENETICS-D-20-01160R1 

Identification of common predisposing loci to hematopoietic cancers in four dog breeds 

Dear Dr Hédan, 

We are pleased to inform you that your manuscript entitled "Identification of common predisposing loci to hematopoietic cancers in four dog breeds" has been formally accepted for publication in PLOS Genetics! Your manuscript is now with our production department and you will be notified of the publication date in due course.

With kind regards,

Alice Ellingham

PLOS Genetics

On behalf of:
